# Response of water use efficiency to summer drought in a boreal Scots pine forest in Finland

Yao Gao[1], Tiina Markkanen[1], Mika Aurela[1], Ivan Mammarella[2], Tea Thum[1], Aki Tsuruta[1], Huiyi Yang[3], Tuula Aalto[1]

[1]Finnish Meteorological Institute, Helsinki, P.O. Box 503, 00101, Finland
[2]Department of Physics, University of Helsinki, Helsinki, P.O. Box 48, 00014, Finland
[3]Institute for Climate and Atmospheric Science, School of Earth and Environment, University of Leeds, Leeds, LS2 9JT, UK

*Correspondence to:* Yao Gao (yao.gao@fmi.fi)

**Abstract.** The influence of drought on plant functioning has received considerable attention in recent years, however our understanding of the response of carbon and water coupling to drought in terrestrial ecosystems still needs to be improved. A severe soil moisture drought occurred in southern Finland in the late summer of 2006. In this study, we investigated the response of water use efficiency to summer drought in a boreal Scots pine forest (*Pinus sylvestris*) at the daily time scale mainly using eddy covariance flux data from the Hyytiälä (southern Finland) flux site. In addition, simulation results from the JSBACH land surface model were evaluated against the observed results. Based on observed data, the ecosystem level water use efficiency (EWUE, the ratio of gross primary production (GPP) to evapotranspiration (ET)) showed a decrease during the severe soil moisture drought, while the inherent water use efficiency (IWUE, a quantity defined as EWUE multiplied with mean daytime vapour pressure deficit (VPD)) increased and the underlying water use efficiency (uWUE, a metric based on IWUE and a simple stomatal model, is the ratio of GPP multiplied with a square root of VPD to ET) was unchanged during the drought. The decrease in EWUE was due to the stronger decline in GPP than in ET. The increase in IWUE was because of the decreased stomatal conductance under increased VPD. The unchanged uWUE indicates that the trade-off between carbon assimilation and transpiration of the boreal Scots pine forest was not disturbed by this drought event at the site. The JSBACH simulation showed declines of both GPP and ET under the severe soil moisture drought, but to a smaller extent compared to the observed GPP and ET. Simulated GPP and ET led to a smaller decrease of EWUE but a larger increase in IWUE because of the severe soil moisture drought in comparison to observations. As in the observations, the simulated uWUE showed no changes in the drought event. The model deficiencies exist mainly due to the lack of the limiting effect of increased VPD on stomatal conductance during the low soil moisture condition. Our study provides deeper understanding of coupling of carbon and water cycles in the boreal ecosystem Scots pine forests and suggests possible improvements to land surface models, which play an important role in the prediction of biosphere-atmosphere feedbacks in the climate system.

**Keywords:** drought response, boreal forests, eddy covariance, water use efficiency, soil moisture drought, land surface model

## 1. Introduction

Terrestrial plants assimilate carbon dioxide ($CO_2$) through photosynthesis accompanied by a loss of water ($H_2O$) in transpiration. Both processes are strongly regulated by local environmental conditions and plant physiology (e.g., stomatal

conductance; $g_s$). Plants protect themselves from excessive water losses (diffusion out of the leaf) under water-limited environments through a reduction of stomatal conductance, which in turn leads to less carbon uptake (diffusion of $CO_2$ into the leaf) and possibly subsequent physiological stress (McDowell et al., 2008; Will et al., 2013).

Soil water deficit can induce a reduction of transpiration (Bréda et al., 1993; Clenciala et al., 1998; Granier et al., 2007; Irvine et al., 1998), and it has been recognized as the main environmental factor limiting plant photosynthesis at global scale (Nemani et al., 2003). Even though the occurrence of drought is low in northern Europe, the summer of 2006 in Finland has been found to be extremely dry and 24.4 % of the 603 forest health observation sites over entire Finland showed drought damage symptoms in visual examination, in comparison to 2–4 % damaged sites in a normal year (Muukkonen et al. 2015). According

to the simulated regional soil moisture, the summer drought in 2006 in southern Finland was the most severe one over the past 30 years (1981-2010), and the spatial distribution of the drought damages has been found to be closely related to the plant available soil moisture (Gao et al., 2016).

Water Use Efficiency (WUE) is a critical metric that quantifies the trade-off between photosynthetic carbon assimilation and

transpiration at the leaf level (Farquhar et al., 1982). WUE can be used to study ecosystem functioning which is in close connection to the global cycles of water, energy and carbon (Keenan et al., 2013). With the use of the eddy covariance technique (EC) and associated data processing, i.e., the derivation of gross primary production (GPP) and evapotranspiration (ET) from measurements of $CO_2$ flux and latent heat flux, WUE can be calculated at the ecosystem scale as the ecosystem level water use efficiency (EWUE), which is the ratio of GPP to ET. EWUE is broadly adopted as a surrogate for the leaf

level WUE in many studies, because more data are available at the ecosystem level than at the leaf level (Arneth et al., 2006; Law et al., 2002; Lloyd et al., 2002).

Reichstein et al. (2007) observed a small decrease in EWUE in the majority of the 11 studied EC sites during the 2003 summer heatwave in Europe. However, their findings are at odds with many models that describe the environmental controls on

stomatal conductance, with increased EWUE predicted during drought periods (Schulze et al., 2005). Many of those models are based on the optimality theory by Cowan and Farquhar (1977) who proposed that plants are able to regulate stomatal conductance in order to maximize WUE. Granier et al. (2008) reported that EWUE increased linearly with soil water deficit duration and intensity at a young beech forest site in north-eastern France. Moreover, EWUE also increased substantially at two forest sites, but not at grassland sites, during the 2011 spring drought in Switzerland (Wolf et al., 2013). However, no

differences in EWUE were shown between abundant- and low-rainfall years at a boreal Scots pine forest site in south-eastern Finland, even though GPP was reduced during low-rainfall years with long-lasting drought periods (Ge et al., 2014). Therefore, the impact of drought on EWUE remains unclear. Beer et al. (2009) concluded that the impact of vapour pressure deficit (VPD) on canopy conductance disturbs responses of both GPP and ET to changing environmental conditions and proposed the ecosystem level inherent water use efficiency (IWUE), which is a quantity defined as EWUE multiplied with mean

daytime VPD. IWUE has been found to increase during short-term moderate drought (Beer et al., 2009). Moreover, based on IWUE and an optimality theory (Cowan and Farquhar, 1977) based stomatal model with the assumptions suggested by Farquhar et al. (1993) and Lloyd and Farquhar (1994), the underlying water use efficiency (uWUE) was introduced to exclude the nonlinear dependence of IWUE on VPD, and the linear relationship between GPP multiplied with a square root of VPD and ET was found at the half-hourly time scale by Zhou et al. (2014). Later on, the appropriateness of uWUE at the daily time

scale was also demonstrated (Zhou et al., 2015).

Given the need to understand and project feedbacks between climate change and plant physiological responses, it is crucial to be able to realistically model the plant controls of stomatal conductance, and photosynthesis and transpiration responses under water stress (Berry et al., 2010; Knauer et al., 2015; Zhou et al., 2013). The various land ecosystem model simulations

highlight the current uncertainty about plant physiology (water use) in response to drought in models (Huang et al., 2015; Jung et al., 2007).

The objectives of this study are (1) to understand the environmental controls on GPP and ET fluxes during a summer drought in boreal Scots pine (*Pinus sylvestris*) forests at a EC flux site in southern Finland, (2) to investigate the drought impact on

WUE metrics, including EWUE, IWUE and uWUE, (3) to evaluate how adequately the JSBACH land surface model (LSM) captures plant responses to changes in environmental variables.

## 2. Data and methods

### 2.1 Study sites

The Hyytiälä flux site is located in southern Finland (61°51′N, 24°17′E, 180 m a.s.l.) at the SMEAR-II (Station for Measuring

Ecosystem-Atmosphere Relations) field measurement station (Hari and Kulmala, 2005). The site is dominated by 55-year-old boreal Scots pine (*Pinus sylvestris*), which is homogeneous about 200 m in all directions from the site and extends to the north for about 1 km (Mammarella et al, 2007). The canopy height of trees is about 13-16 m and the mean all-sided leaf area index (LAI) is 6 $m^2/m^2$. The soil at the site is Haplic podzol on glacial till (FAO-UNESCO, 1990). The 30-year (1961-1990) averaged annual mean air temperature is 2.9 °C and precipitation is 709 mm at the site (Vesala et al., 2005). Those details

about the site are listed in Table 1. The ground vegetation consists mainly of blueberry (*Vaccinium myrtillus*), lingonberry (*Vaccinium vitis-idaea*), feather moss (*Pleurozium schreberi*) and other bryophytes (Kolari et al., 2009). We analysed the summer (June-August) from an 11-year period (1999-2009) according to data availability.

### 2.2 Flux measurement and data processing

Ecosystem carbon and water fluxes at the site were measured with the micrometeorological EC method. Turbulent fluxes were calculated as half-hourly averages following standard methodology (Aubinet et al., 2012) with EddyUH software (Mammarella et al., 2016). The vertical $CO_2$ flux was obtained as the covariance of high-frequency (10 Hz) observations of vertical wind speed and the $CO_2$ concentration (Baldocchi, 2003). The $CO_2$ flux was corrected for storage change to obtain net ecosystem $CO_2$ exchange (NEE), which was then partitioned into total ecosystem respiration (TER) and GPP according

to Kolari et al. (2009). Data quality of 30 min values of NEE and latent heat flux (LE) was ensured by excluding records with low turbulent mixing (friction velocity below 0.25 m/s) as described in Markkanen et al. (2001), Mammarella et al (2007) and Ilvesniemi et al. (2010). TER was modelled using an exponential equation with temperature at a depth of 2 cm in the soil organic layer as the explanatory factor. The value of GPP was then directly derived as residual from the measured NEE. When NEE was missing, GPP was gap-filled according to Kolari et al. (2009). LE was gap-filled using a linear regression against

net radiation in a moving window of 5 days, and then ET was inferred from LE.

In addition to the EC measurements, a set of supporting meteorological variables were adopted as half-hourly averages; incoming shortwave radiation ($R_s$) and longwave radiation, air temperature ($T_a$), atmospheric humidity, precipitation were used as meteorological forcing for the site level simulation. The soil moisture was monitored at 1-hour intervals by the Time Domain Reflectometry (TDR) method (Tektronix 1502 C cable radar, Tektronix Inc., Redmond, USA). Three layers of mineral soil (0 to 5, 5 to 23 and 23 to 60 cm) were measured, as well as the organic layer on the top (-4 to 0 cm). In this study, soil moisture at the two lower levels of mineral soil (5 to 23 and 23 to 60 cm) at Hyytiälä was averaged over a day to represent daily soil moisture dynamics in the root zone at the site. The reason to exclude layer 1 soil moisture is that it is too sensitive to temperature and precipitation variations.

The half-hourly data of GPP and ET, as well as meteorological variables were averaged over the selected time periods in a day. Prior to averaging, rainy days and a number of dry days after the rainy days were firstly excluded from the data. The number of excluded dry days was determined by the ratio of daily precipitation to potential evapotranspiration (PET). When precipitation was smaller than PET, no dry day after rainy day was excluded. When precipitation was equal or larger than twice of PET, two dry days following the rainy day were excluded. Additionally, when precipitation was larger than PET but with the ratio less than two, one dry day after the rainy day was excluded. PET was calculated using the Penman-Monteith equation and the 'Evapotranspiration' Package in R software was used (Guo et al., 2016). Second, in order to capture the daily time periods of effective photosynthesis, only half-hourly data with $R_s$ larger than 100 W/m$^2$ were selected. Finally, the half-hourly data of $R_s$, VPD, and $T_a$ were also averaged over the selected time periods to get their daytime mean values respective to the GPP and ET data. The same data processing method was used for the simulation results.

## 2.3 JSBACH land surface model

JSBACH (Raddatz et al., 2007; Reick et al., 2013) is the LSM of the Max Planck Institute for Meteorology Earth System Model (MPI–ESM) (Roeckner et al., 1996; Stevens et al., 2013). The land physics of JSBACH mainly follow those of the global atmosphere circulation model ECHAM5 (Roeckner et al., 2003), and the biogeochemical components are mostly taken from the biosphere model BETHY (Knorr, 2000). In JSBACH, land vegetation cover is described as plant functional types (PFTs) and a set of properties (e.g., maximum LAI, albedo) is attributed to each PFT with respect to the processes that are accounted for by JSBACH. The phenology model (Logistic Growth Phenology; LoGro-P) of JSBACH simulates the LAI dynamics to compute photosynthetic production (Böttcher et al., 2016). The models of Farquhar et al. (1980) and Collatz et al. (1992) are used for photosynthesis of C3 and C4 plants, respectively. A five layer soil hydrology scheme was implemented in JSBACH by Hagemann and Stacke (2015). Gao et al. (2016) has demonstrated that JSBACH with its five-layer soil hydrology scheme is able to capture the soil moisture dynamics at sites and in the regional scale of Finland.

### 2.3.1 The stomatal conductance model in JSBACH

The current version of the stomatal conductance model in JSBACH considers the limitation from soil water availability on stomatal conductance ($g_s$), which further impacts on carbon assimilation and transpiration.

Firstly, the net assimilation rate ($A_n$ [mol m$^{-2}$ s$^{-1}$]) and $g_s$ [mol m$^{-2}$ s$^{-1}$] are calculated for without water limitation as the unstressed net assimilation rate ($A_{n,pot}$ [mol m$^{-2}$ s$^{-1}$]) and the unstressed stomatal conductance ($g_{s,pot}$ [mol m$^{-2}$ s$^{-1}$]). The $A_{n,pot}$ is calculated using the photosynthesis model in JSBACH, for which the intercellular $CO_2$ concentration under unstressed

condition ($C_{i,pot}$ [mol mol$^{-1}$]) is needed. The $C_{i,pot}$ is prescribed using the atmospheric $CO_2$ concentration ($C_a$ [mol mol$^{-1}$]), where $C_{i,pot} = 0.87C_a$ for C3 plants and $C_{i,pot} = 0.67C_a$ for C4 plants (Knorr, 2000). After the $A_{n,pot}$ is determined, the $g_{s,pot}$ is derived using the following equation:

$$g_{s,pot} = \frac{1.6A_{n,pot}}{C_a - C_{i,pot}} \tag{1}$$

Then, an empirical water stress factor, which is a function of volumetric soil moisture, is used to derive $g_s$ [mol m$^{-2}$ s$^{-1}$] from $g_{s,pot}$ as follows:

$$g_s = \beta g_{s,pot} \tag{2}$$

where

$$\beta = \begin{cases} 1 & \theta \geq \theta_{crit} \\ \frac{\theta - \theta_{wilt}}{\theta_{crit} - \theta_{wilt}} & \theta_{wilt} < \theta < \theta_{crit} \\ 0 & \theta \leq \theta_{wilt} \end{cases} \tag{3}$$

herein, $\theta$ [m$^3$ m$^{-3}$] is the volumetric soil moisture, $\theta_{crit}$ [m$^3$ m$^{-3}$] is the critical point and $\theta_{wilt}$ [m$^3$ m$^{-3}$] is the permanent wilting point.

Finally, the intercellular $CO_2$ concentration ($C_i$) and $A_n$ are resolved using $g_s$. The canopy conductance ($G_c$ [mol m$^{-2}$ s$^{-1}$]) and canopy-scale $A_n$ are integrated over the leaf area. Unlike the BETHY approach (Knorr, 2000), the control of $g_s$ in JSBACH does not include the influence of atmospheric humidity.

## 2.4 Site level simulation by JSBACH

For the site simulation, JSBACH was forced with the half-hourly local meteorological observations. Based on the site-specific information, PFT was assigned as evergreen needleleaf forest and the soil type was set as loamy sand in JSBACH. The modelled LAI reached values close to the observed LAI when the parameter maximum LAI was set to 16 m$^2$/m$^2$. Also, the maximum carboxylation rate (Jmax) and maximum electron transport rate (Vmax) at 25 °C were adjusted, for the simulated
GPP to match the magnitude of the observed GPP. The Vmax was set to be 37.5, and the Jmax was 71.3. The soil depth and root depth at the site were derived from maps for the regional JSBACH simulation presented in Gao et al. (2016) (see also Hagemann and Stacke, 2015). Those parameter settings in the JSBACH site level simulation for the site are listed in Table 1. Prior to the actual simulations, a 30-year spin-up run was conducted by cycling meteorological forcing that was used for the actual simulation to obtain equilibrium for soil water and soil heat balances.

## 2.5 Soil Moisture Index (SMI)

In this study, the soil moisture dynamics are represented by SMI (also referred to as Relative Extractable Water – REW), which has been demonstrated to represent summer drought in boreal forests in Finland (Gao et al., 2016). The SMI describes the ratio of plant available soil moisture to the maximum volume of water available to plants in the soil (Betts, 2004; Seneviratne et al., 2010):

$$SMI = (\theta - \theta_{WILT})/(\theta_{FC} - \theta_{WILT}), \tag{4}$$

where $\theta$ is the volumetric soil moisture [m$^3$ H$_2$O m$^{-3}$], $\theta_{FC}$ is the field capacity [m$^3$ H$_2$O m$^{-3}$] and $\theta_{WILT}$ is the permanent wilting point [m$^3$ H$_2$O m$^{-3}$]. When $\theta$ exceeds $\theta_{FC}$, soil water cannot be retained against gravitational drainage, while below $\theta_{WILT}$, the

soil water is strongly held by the soil matrix and cannot be extracted by plants (Hillel, 1998). In this study, soil moisture conditions were classified to five groups according to SMI values with an interval of 0.2: very dry: $0 \leq SMI < 0.2$, moderate dry: $0.2 \leq SMI < 0.4$, mid-range: $0.4 \leq SMI < 0.6$, moderate wet: $0.6 \leq SMI < 0.8$, very wet: $0.8 \leq SMI < 1$.

From simulations, we used the average of the second layer (layer-2; 6.5–31.9cm) and the third layer (layer-3; 31.9–123.2 cm) soil moisture together with model soil parameters to determine the simulated SMI for Hyytiälä, for the aim to correspond with the observed SMI that calculated with measured soil moisture at the two lower levels of mineral soil at the site. The layer 1 soil moisture was excluded in determining both simulated and observed SMIs because it is too sensitive to temperature and precipitation variations. For the observed SMI, the measured soil parameters derived based on water retention curves determined from soil samples taken at the site were adopted (i.e., volumetric soil moisture at saturation ($\theta_{SAT}$) = 0.50 $m^3$ $H_2O$ $m^{-3}$, $\theta_{FC}$ = 0.30 $m^3$ $H_2O$ $m^{-3}$ and $\theta_{WILT}$ = 0.08 $m^3$ $H_2O$ $m^{-3}$). As $\theta_{FC}$ acts as a proxy for $\theta_{SAT}$ in the five layer soil hydrology scheme in JSBACH (Hagemann and Stacke, 2015), $\theta_{SAT}$ was used instead of $\theta_{FC}$ for consistency when calculating SMI based on the observed soil moisture data.

**2.6 Ecosystem Water Use Efficiency (EWUE), Inherent Water Use Efficiency (IWUE) and Underlying Water Use Efficiency (uWUE)**

The ecosystem level water use efficiency is calculated as,

$$EWUE = GPP/ET, \tag{5}$$

IWUE is defined as EWUE multiplied by daytime mean VPD in Beer et al. (2009),

$$IWUE = GPP \times VPD/ET, \tag{6}$$

uWUE is derived based on IWUE and an optimality theory (Cowan and Farquhar, 1977) based stomatal model with the assumptions suggested by Farquhar et al. (1993) and Lloyd and Farquhar (1994) in Zhou et al. (2014). The formulation of uWUE is,

$$uWUE = GPP \times VPD^{0.5}/ET \tag{7}$$

From EC data EWUE and IWUE can only be calculated with ET, which, in addition to transpiration, contains evaporation of water intercepted by surfaces and soil evaporation. However, process-based ecosystem models do resolve evaporation and transpiration which together compose ET. Therefore, transpiration-based EWUE, IWUE and uWUE can also be calculated using simulated transpiration instead of ET in those equations.

**3. Results**

**3.1 Soil moisture drought at Hyytiälä in 2006**

In the summer of 2006, a period with evidently lower SMI values (< 0.2) than in any other year during the 11-year time series was shown (Fig. 1 (a)). According to the in situ observation, in the summer of 2006, there were 37 consecutive days (23 July to 28 August) with SMI lower than 0.2, and 17 consecutive days (1 August to 17 August) with SMI lower than 0.15. The observed SMI reached its minimum of 0.115 on 16 August 2006. The simulated SMI was generally smaller than the observed

SMI in the summer of 2006, showing 42 consecutive days (17 July to 27 August) with SMI lower than 0.2, and 33 consecutive days (26 July to 27 August) with SMI lower than 0.15. The lowest SMI from simulation was 0.052 on August 15$^{th}$. The simulated SMI agreed well with in situ observed SMI over the 11-year study period, with a correlation coefficient of 0.63 and a root-mean-square error (RMSE) of 0.23. However, the simulated SMI showed larger amplitude and a faster response to changes in climate conditions in comparison to the observed SMI. Nevertheless, a very good correlation coefficient of 0.97 between simulated and observed SMIs were found for year 2006 (Fig. 1 (b)), despite the simulated SMI being systematically lower than the observed SMI (RMSE = 0.12).

Concurrently with the low soil moisture, a high $T_a$ anomaly was observed in August 2006 (Fig. 1 (c)). In all the days in August 2006, the daily mean in situ $T_a$ was higher than the 11-year averaged daily mean $T_a$. The monthly mean $T_a$ in August 2006 (18.1 ± 1.9 °C) was 3.1 °C higher than that of the 11-year average (15.0 ± 1.63 °C). Also, the daily mean VPD in August 2006 was higher than the 11-year averaged daily mean VPD in August in general (not shown), except on the days with precipitation. Especially, the mean value of the daily mean VPD in the period from 31 July to 16 August (1.067 ± 0.361 kPa) was substantially higher than the mean of the 11-year averaged daily mean VPD over this period (0.582 ± 0.200 kPa). The biggest difference between the daily mean VPD and the 11-year averaged daily mean VPD reached 1.054 kPa on August 5$^{th}$ that was the day with highest $T_a$ in August 2006. The daily mean $R_s$ in the summer of 2006 was overall higher than the 11-year averaged daily mean $R_s$, with the monthly mean values by 15.4%, 31.2% and 21.4 % higher in June, July and August respectively.

The precipitation events have strong impact on the temporal pattern of SMI. The cumulative in situ precipitation of 34 mm in July 2006 was the lowest during the 11-year study period with the July average of 91 ± 31 mm. In contrast, the highest total precipitation in July was in 2007, reaching 146 mm. The cumulative precipitation of 48 mm in August 2006 was not as low as in July when compared to the 11-year average of 71 ± 43 mm. However, the lack of precipitation since the end of July led to the continuous drop of SMI till mid August 2006, followed by a small increase in soil moisture after a light precipitation event. The SMI increased to be above 0.2 in the end of August with a heavy precipitation event exceeding 25 mm in one day. Moreover, the precipitation in June 2006 was also less than the 11-year average (45 vs. 70 ± 24 mm) and temporally unevenly distributed, with only a small amount in the beginning of June and a large amount in the end of June. Therefore, there was a continuous decrease of soil moisture from the beginning of June and an abrupt increase in SMI of more than 0.1 in the end of June.

**3.2 The relationship of GPP to ET categorized by environmental variables**

In general, the daytime averaged GPP and ET from observations at Hyytiälä showed a non-linear relationship (Fig. 2 (a)). When categorized according to environmental variables, there is a group of data under the very dry soil moisture condition (encircled with a dashed line in Fig. 2 (a)) showing GPP values lower than other days. The ET values of this group are also located in the lower end, but just partly lower than ET values on other days. It is found that the days in this group are with SMI smaller than 0.15. Moreover, there are only two days with SMI values smaller than 0.15 that are not included in the encircled group due to their slightly higher GPP values. Most of the days in the group have high daytime mean $T_a$ (18 to 24 °C), sufficient daytime mean $R_s$ (mostly above 300 W/m$^2$), and relatively high daytime mean VPD (above 1 kPa).

The non-linear relationship between the daytime averaged GPP and ET was also found in the JSBACH simulated result (Fig. 2 (b)). The decline of both GPP and ET during low SMI was captured by the model. However, under the very low soil moisture

condition (SMI < 0.15) during the summer drought in 2006, the model simulated much less reduction of GPP, while the ET decreased to be lower than the observation in few days. The non-linear relationship between simulated daytime averaged GPP and transpiration (Fig. S1 in supplementary) is similar to the relationship between simulated daytime averaged GPP and ET, which demonstrates that transpiration composes a large fraction of ET at daytime at the site, especially under soil water stress. Except the drought events, GPP and ET both increased with increasing $R_s$ and VPD in the simulation, which was more evident than in the observational data.

### 3.3 Response of GPP and ET to environmental variables categorized by SMI

The dependence of GPP and ET on environmental variables was further investigated for different SMI ranges (Fig. 3). The exclusion of the night time and the days affected by rain (see details in section 2.2) removed also the small values of GPP and ET. Linear regressions were fitted between GPP (ET) and environmental variables for each soil moisture group to emphasize the deviating differences of dependence of GPP (ET) on environmental variables under different soil moisture conditions. The regression parameters, correlation coefficient and statistical significance are summarized in Table S1 in the Supplementary Material.

The very dry soil ($0 \leq$ SMI < 0.2) led to a response of observed daytime mean GPP and ET to daytime mean $R_s$, $T_a$ and VPD that deviated considerably from the responses of the daily mean SMI values greater than 0.2 (Fig.3 (a)). Under the very dry soil moisture condition, GPP decreased with the declining SMI with a high correlation of 0.79, whereas the other SMI groups showed more scattered relationship between GPP and SMI. Different from the other SMI groups, GPP was the most negatively correlated with $T_a$ and VPD under the very dry soil moisture condition. Moreover, the group with SMI values less than 0.2 displayed lower GPP values (on average 97.6 µg C $m^{-2}s^{-1}$) than the other groups (on average 151 µg C $m^{-2}s^{-1}$). The response patterns of the observed ET to environmental variables were similar to those of GPP. As with GPP, the group under the very dry soil moisture condition deviated strongly from the other SMI groups. However, the decrease in ET under severe soil moisture drought was not as pronounced as in GPP.

For the simulated GPP and ET too, the group under the very dry soil moisture condition deviated from the other SMI groups, but not to the same extent as that in the observed GPP and ET. Under other soil moisture conditions (SMI >0.2), the simulated GPP had stronger positive linear relationships with daytime mean $R_s$, $T_a$ and VPD than the observed GPP. Compared to the observed ET, some differences existed in the response of the simulated ET to environmental variables. First, the dependence of simulated ET on $R_s$ tended to be more linear than the observed ET and $R_s$ relationship. Second, unlike observed ET, the simulated ET increased concomitantly with VPD at high VPD. Nevertheless, simulated ET of the group under severe soil moisture drought deviated strongly from the other SMI groups, but to a less extent than observed ET.

### 3.4 Soil moisture drought impacts on EWUE, IWUE and uWUE

From the observation, the decrease in GPP was much stronger than the decrease in ET during the soil moisture drought, which resulted in largely decreased EWUE that reached the recorded minimum during the severe soil moisture drought (Fig. 2 and Fig. S2 in Supplementary Material). In contrast to EWUE, IWUE increased from 3.25 µg C kPa /mg $H_2O$ (the mean value for the days with SMI equal or larger than 0.2) to 3.93 µg C kPa /mg $H_2O$ (the mean value for the days with SMI smaller than 0.2), and uWUE did not change under the severe soil moisture drought at Hyytiälä (Fig. 4 (a)). The simulated EWUE decreased

less and the simulated IWUE increased more (from 3.62 µg C kPa /mg $H_2O$ to 5.17 µg C kPa /mg $H_2O$) than the observation, which is mainly because of a smaller decrease of the simulated GPP than its observed counterpart during the soil moisture
drought (Fig. 4 (b)). The simulated uWUE remained insensitive to the severe soil moisture drought. In addition, the transpiration based EWUE, IWUE and uWUE (Fig. S3 in Supplementary Material) showed similar results to those three metrics calculated with ET.

## 4. Discussion

**4.1 Drought impacts on GPP and ET**

Both GPP and ET were suppressed when there was the severe soil moisture drought in the summer of 2006 at Hyytiälä. In addition, the response of GPP and ET to the changes in environmental variables under severe water stress differed from those under other soil moisture conditions. The dominant reason is that low soil moisture leads to stomatal regulation of the plants, which limits plant carbon assimilation and transpiration. The decreased ET due to soil moisture drought may increase
atmospheric VPD, which could in turn intensify stomatal closure (Eamus et al., 2013; Jarvis, 1976). Moreover, the GPP and ET were decoupled and EWUE decreased due to the soil moisture drought. Different to EWUE, IWUE increased but uWUE showed no changes during the severe soil moisture drought at Hyytiälä. IWUE depends on the difference between ambient partial pressure of $CO_2$ ($C_a$) and a weighted average of inner leaf partial pressure of $CO_2$ ($C_i$) through the canopy within the tower footprint (Beer et al., 2009). It has been shown that the term ($1-C_i/C_a$) increases as VPD increases (Wong et al., 1979).
Thus, the increase in IWUE during drought was a result of decreased stomatal conductance due to increased VPD. The uWUE was formulated to be more independent of a varying VPD than IWUE. According to Xie et al. (2016), both IWUE and uWUE at a flux site increased and reached their maximum values over long-term during a severe drought in central and southern China in the summer of 2013. In this work, the unchanged uWUE during this drought event demonstrate that the trade-off between carbon assimilation and transpiration of the boreal Scots pine forest was not disturbed by drought at the study site,
even though the stomatal conductance decreased.

**4.2 Differences between observations and site simulations**

The model showed the limitations on GPP and ET under the very dry soil moisture condition ($0 \leq SMI < 0.2$) at Hyytiälä. However, the discrepancies in response between observed and simulated GPP and ET to changing environmental variables were obvious. This is because the formulation for stomatal conductance in JSBACH does not include a response to air
humidity, and therefore the stomatal conductance in JSBACH is insensitive to atmospheric VPD (Knauer et al., 2015). In Knauer et al. (2015), Ball-Berry model (Ball et al., 1987) has been found to be the best among a few stomatal conductance models in its response to atmospheric drought under non-limited soil moisture conditions. In reality, low soil moisture and high $T_a$ during drought are closely coupled with high atmospheric VPD. Our results indicate that the combined effects of soil moisture and atmospheric drought on stomatal conductance have to be taken into account. Moreover, model performance
could be improved through the inclusion of non-stomatal limitations on plant photosynthesis, which have been considered to be important for the simulation of short-term plant responses to drought (Egea et al., 2011; Manzoni et al., 2011; Zhou et al., 2013). However, JSBACH is being continuously developed and the effect of soil water stress is to be accounted according to Egea et al., 2011 for both stomatal and non-stomatal processes, affecting both conductance and photosynthesis parameters.

Moreover, when comparing results from the EC data and simulations, it should be kept in mind that the EC method has its uncertainties. Due to the stochastic nature of the turbulent flow, there is always a random error component in the observations. In addition, imperfect spectral corrections and gap-filling procedures as well as calibration problems may be sources of systematic errors (Richardson et al., 2012; Wilson et al., 2002). The uncertainty of EC flux data is typically 20-30% for annual carbon budget (Aubinet et al., 2012; Baldocchi, 2003). Nevertheless, the uncertainties of the GPP and ET estimated from EC
measurements are likely to have negligible impacts on our findings of the three WUE metrics, as the same data with the same uncertainties were used.

## 5. Conclusions

In this study, the impact of the severe soil moisture drought in the summer of 2006 on the water use efficiency of a boreal Scots pine forest ecosystem at Hyytiälä flux site in southern Finland was investigated using both ground-based observations
from a flux tower and the site-level simulation by the JSBACH land surface model. The soil moisture index (SMI) was used to indicate the soil moisture condition at the site. Finland is a high-latitude country and drought is uncommon. Nevertheless, the summer drought in 2006 caused severe forest damage in southern Finland (Muukkonen et al., 2015). The SMI calculated from regional soil moisture simulations over the past 30 years (1981-2010) indicated that such extreme drought affecting forest health was rare in Finland, and the summer drought in 2006 in southern Finland was the most
severe one in the 30-year study period (Gao et al., 2016). According to climate scenarios, regardless of the anticipated increase of precipitation, a modest drying of soil is foreseen in the Northern-Europe during the 21st century because of intensifying evapotranspiration (Ruosteenoja et al., 2017).

The impacts from the severe soil moisture drought on plant functioning at the site were clearly seen in the gross primary
production (GPP) and evapotranspiration (ET). From both observation and simulation results, the GPP and ET reached the recorded minimums during the drought event. The ecosystem water use efficiency (EWUE) decreased, whereas the inherent water use efficiency (IWUE) increased and the underlying water use efficiency (uWUE) was unchanged during the severe soil moisture drought at the site. The EWUE is very sensitive to the daily changes of GPP and ET. The increase in IWUE during drought was due to the decreased stomatal conductance of plants under increased VPD. The unchanged uWUE
indicates that the carbon assimilation and transpiration coupling of the boreal Scots pine forest was not disturbed by the drought event at this site, although the stomatal conductance of plants decreased.

The simulated response in plant functioning to the severe soil moisture drought predicted by JSBACH was weaker than those in the observed dataset, even though the strong limitation on GPP and ET through stomatal closure were seen at the very dry
soil moisture condition ($0 \leq SMI < 0.2$) as in the observed data. The differences between the observed and the model results suggest that, in order to adequately simulate effects of drought on plant functioning, the combined effects of atmospheric and soil moisture drought on stomatal conductance have to be included in the stomatal conductance model in JSBACH. Moreover, inclusion of non-stomatal limitations on photosynthesis during drought, e.g., reduced mesophyll conductance or carboxylation capacity, may additionally improve the model results (Keenan et al., 2010).

This study gives a view of the response of water use efficiency to a summer drought event in a boreal Scots pine forest in

Finland, and further suggests that improving our knowledge of ecosystem processes in land surface models are of great importance when estimating biosphere-atmosphere feedbacks of terrestrial ecosystems under climate change.

## Acknowledgements

We would like to thank the Academy of Finland Centre of Excellence (272041), Academy of Finland (266803), OPTICA (295874), CARB-ARC (285630), ICOS Finland (281255), ICOS-ERIC (281250), MONIMET (LIFE07 ENV/FIN/000133, LIFE12 ENV/FIN/000409), NCoE eSTICC (57001) and EU FP7 EMBRACE (282672) projects for support. The authors acknowledge MPI-MET and MPI-BGC for fruitful co-operation regarding use of JSBACH model.

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

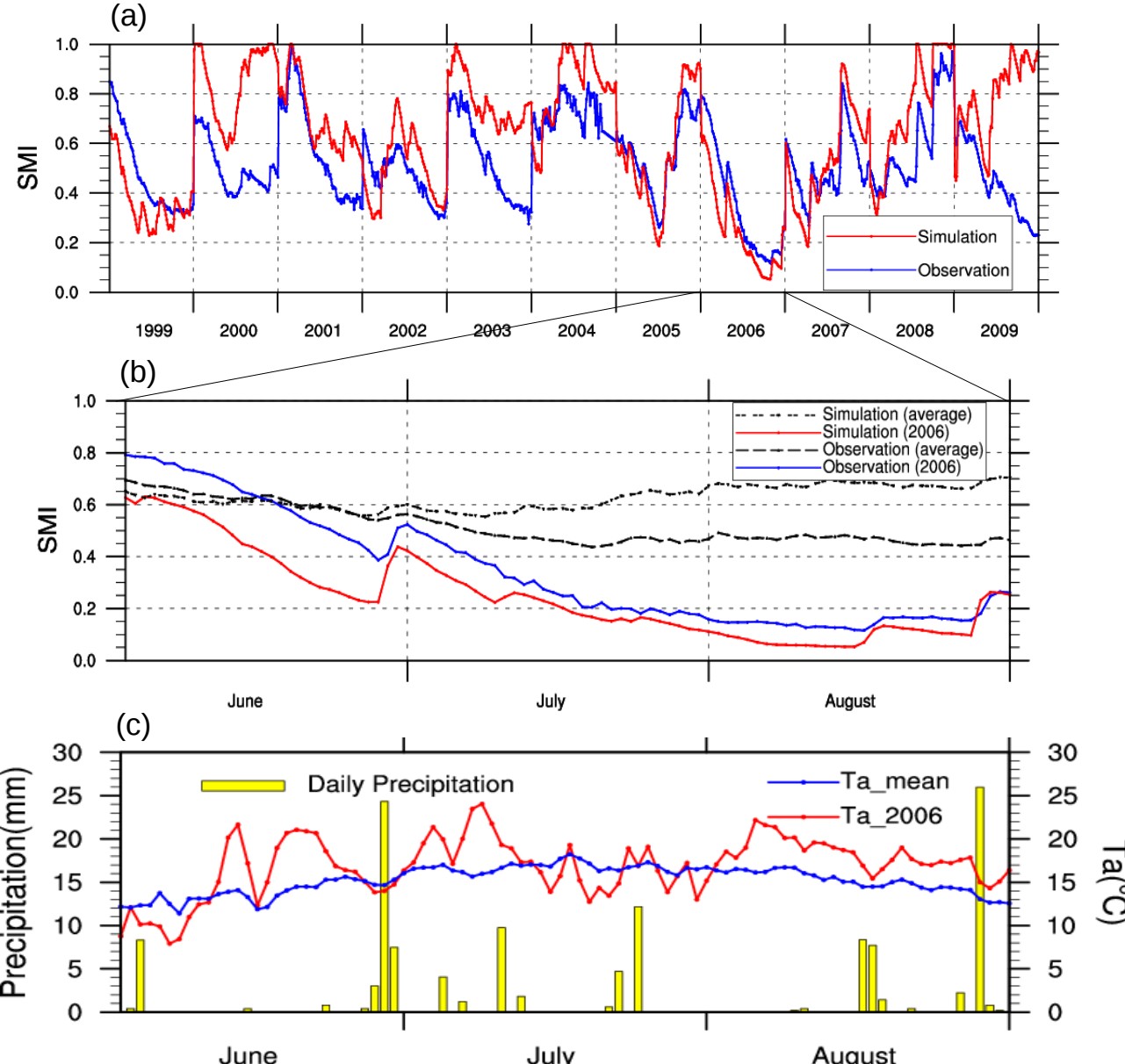

Figure 1: (a) Daily mean soil moisture index (SMI) at Hyytiälä from observation and the JSBACH simulation for the summer months (June, July, August) in the 11-year study period (from 1999 to 2009). (b) Daily mean SMI at Hyytiälä from observation and the JSBACH simulation for the summer months in 2006; The two black dashed lines represent the averaged daily SMI in the summer months over the 11-year study period from observation and the JSBACH simulation. (c) Daily mean air temperature ($T_a$) in the summer months of 2006 and the averaged daily mean $T_a$ in the summer months over the 11-year study period at Hyytiälä from observation, meanwhile, the daily precipitation amount in 2006 is shown as bar plot.

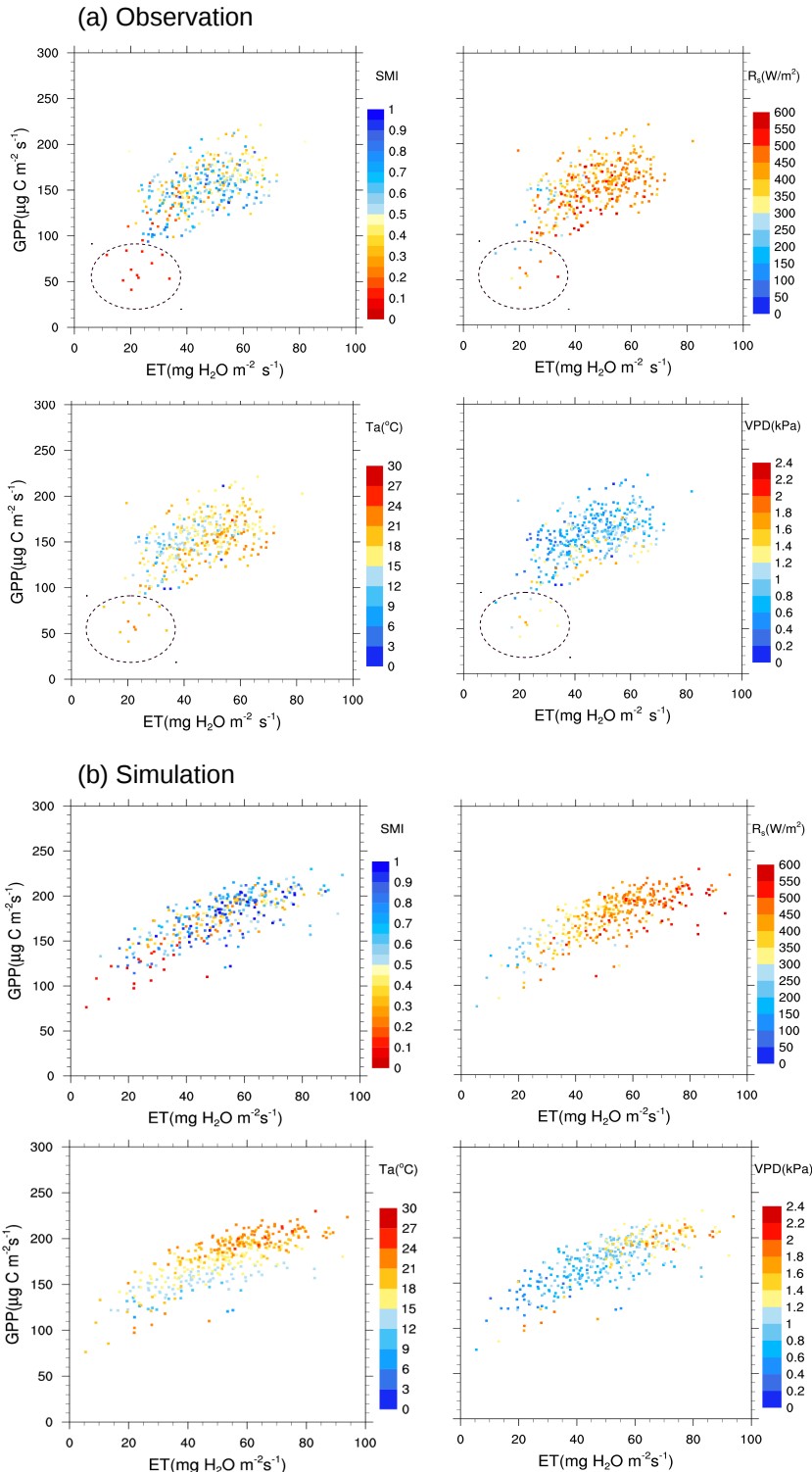

**Figure 2: Relationship between the daytime averaged gross primary production (GPP in μg C m$^{-2}$s$^{-1}$) and evapotranspiration (ET in mg H$_2$O m$^{-2}$ s$^{-1}$) at Hyytiälä in the summer months (June, July, August) of the 11-year study period (from 1999 to 2009) from (a) observation and (b) the JSBACH simulation. Data are categorized according to daily mean soil moisture index (SMI), daytime mean incoming shortwave radiation (R$_s$), daytime mean air temperature (T$_a$) and daytime mean vapour pressure deficit (VPD), respectively. In the observation, the group of data under the very dry soil moisture condition showing GPP values lower than other days is marked with dashed circle.**


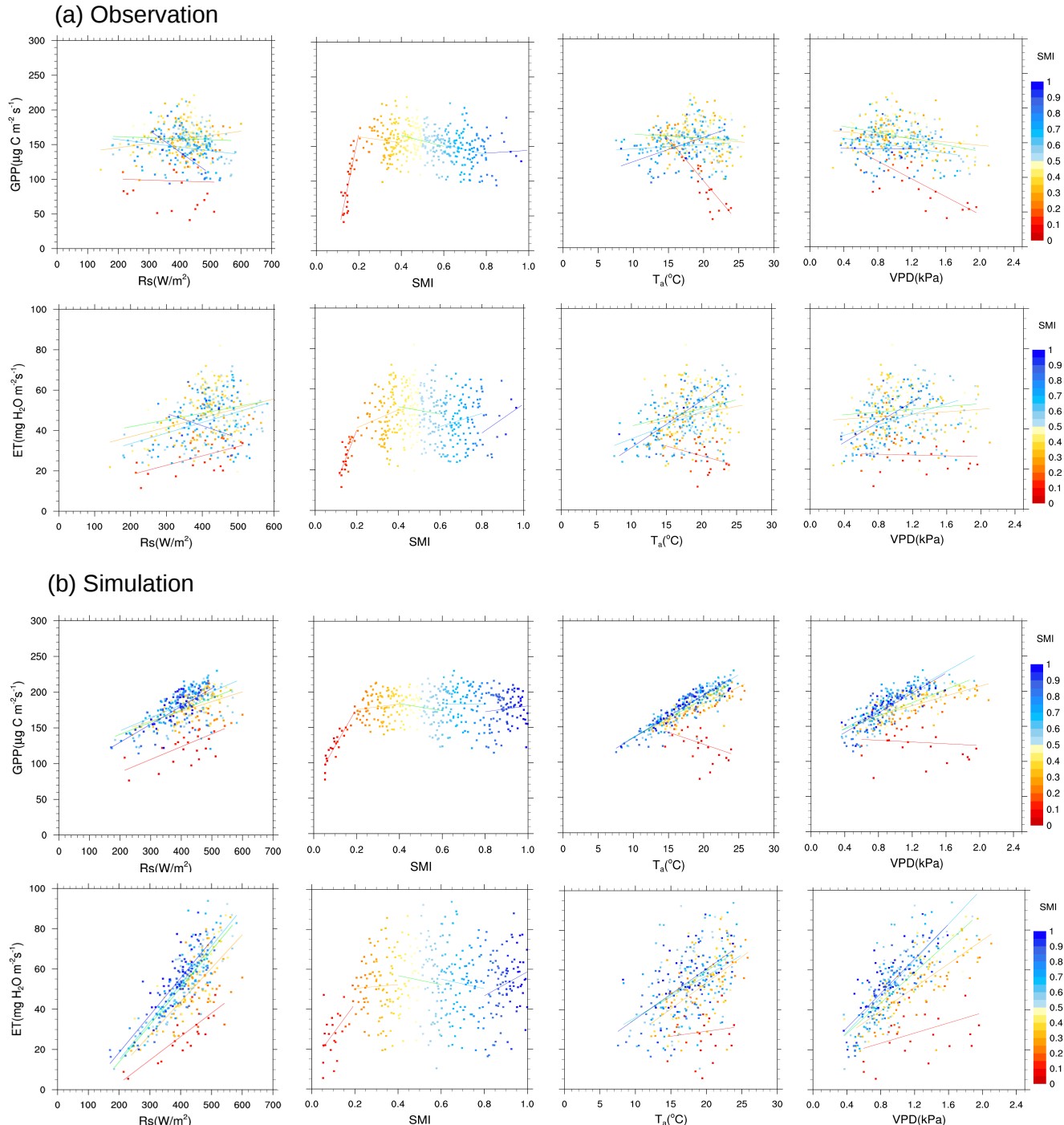


**Figure 3: Response of daytime mean gross primary production (GPP in µg C m⁻²s⁻¹) and evapotranspiration (ET in mg H₂O m⁻² s⁻¹) to daytime mean incoming shortwave radiation (R$_s$), daytime mean air temperature (T$_a$), daytime mean vapour pressure deficit (VPD), and daily mean soil moisture index (SMI) at Hyytiälä, categorized by daily mean soil moisture index (SMI) in the summer months (June, July, August) of the 11-year study period (from 1999 to 2009)**

**from (a) observation and (b) the JSBACH simulation. The regression lines are fitted for the five SMI groups (very dry: 0 ≤ SMI < 0.2, moderate dry: 0.2 ≤ SMI < 0.4, mid-range: 0.4 ≤ SMI < 0.6, moderate wet: 0.6 ≤ SMI < 0.8, very wet: 0.8 ≤ SMI <1).**

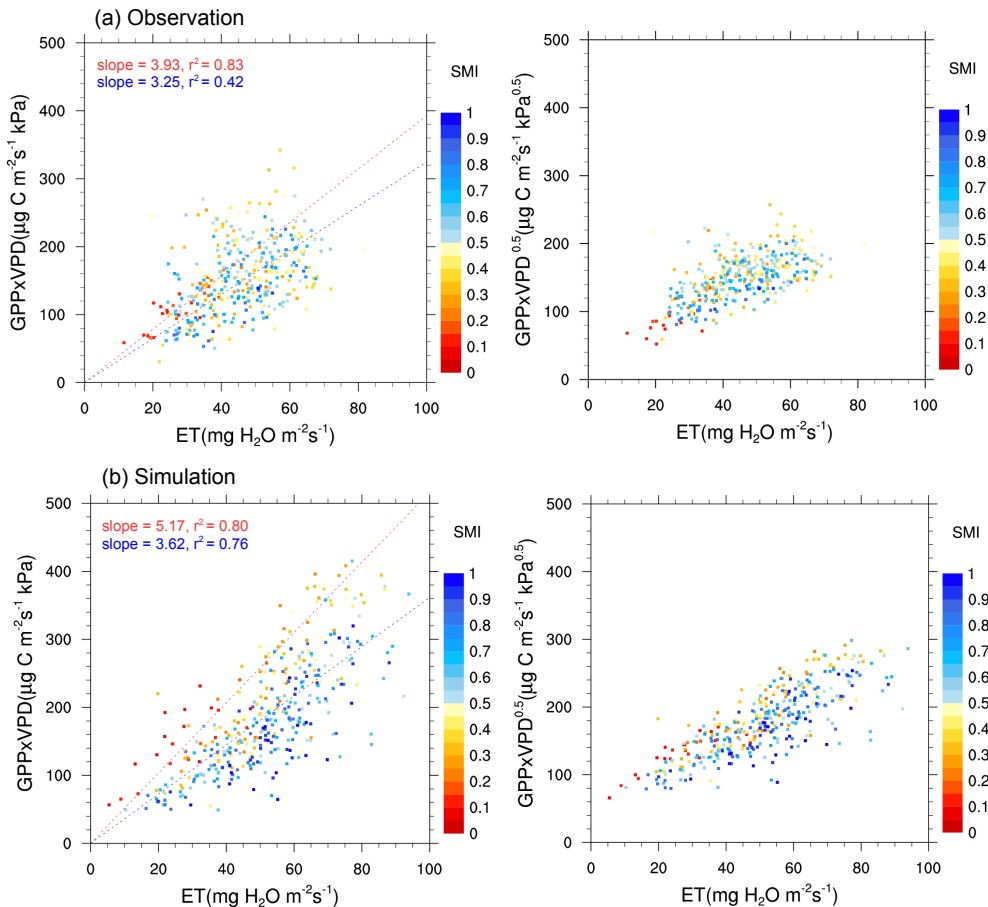


**Figure 4: The dependence of the product of daytime mean gross primary production (GPP in µg C m$^{-2}$s$^{-1}$) and daytime mean vapour pressure deficit (VPD) on evapotranspiration (ET in mg H$_2$O m$^{-2}$ s$^{-1}$) (i.e., GPP×VPD/ET, which represents the inherent water use efficiency (IWUE)), and the dependence of the production of GPP and the square root of VPD on ET (i.e., GPP×VPD$^{0.5}$/ET, which represents the underlying water use efficiency (uWUE)) in the summer**

**months (June, July, August) of the 11-year study period (from 1999 to 2009) from (a) observation and (b) the JSBACH simulation. Data are categorized according to daily mean soil moisture index (SMI). The fitted lines for the dependence of the product of GPP and VPD on ET are for the data under SMI < 0.2 (red line) and the data under 0.2 ≤ SMI <1 (blue line), respectively; both fittings are statistically significant (p-value < 0.05). No lines were fitted for the dependence of the production of GPP and the square root of VPD on ET, as the data under SMI < 0.2 and data under**

**0.2 ≤ SMI <1 are more converged in a line in comparison to the dependence of the product of GPP and VPD on ET.**


**Table 1: Key characteristics relevant to this study from observation and the parameter settings in the JSBACH site level simulation at Hyytiälä site.**

| | | | | | | Observation | | | |
|---|---|---|---|---|---|---|---|---|---|
| Site | Location | Vegetation type | LAI $(m^2/m^2)$ (all-sided, annual) | Canopy height (m) | Measurement height (m) | Annual mean air temperature (∘C) and precipitation (mm) (30-year averages) | Soil type | Analysed measurement depth of soil moisture (cm) | References |
| Hyytiälä | 61°51'N, 24°17'E | Scots pine | 6 | 13-16 | 23 | 2.9; 709 | Mineral (Haplic podzol) | 5 to 23; 23 to 60 | Markkanen et al.(2001); Vesala et al. (2005) |

| | | | | | Settings in JSBACH | | | | |
|---|---|---|---|---|---|---|---|---|---|
| Site | PFT | Maximum LAI $(m^2/m^2)$ | Maximum electron transport rate (Vmax) at 25 ∘C | Maximum carboxylation rate at 25 ∘C | Soil type | Analysed depth of soil moisture (cm) | Soil depth (m) | Root depth (m) | |
| Hyytiälä | Evergreen needleleaf forest | 16 | 37.5 | 71.3 | Loamy sand | Average of layer-2 (6.5–31.9) and layer-3 (31.9–123.2) | 5.416 | 1.265 | |