# Peer review of "Response of water use efficiency to summer drought in a boreal Scots pine forest in Finland"

_Biogeosciences, 2016_

## Referee Comment (RC1) · Anonymous Referee #1 · 20 Jun 2016

General comments

This paper deals with an important issue relating the water and carbon cycles on the ecosystem level. The authors used two long-term datasets and derived interesting results about water use efficiency.

The introduction is good, explaining well the problem and citing many recent references. The methodology describes well the sites characteristics and the methods used.

Finally, the text is well written and the quality of figures is good.

Specific comments

Abstract Lines 16-17: It would be better to have a sentence or two explaining how

EWUE and IWUE were defined, since different definitions might be used such as NEE/ET for EWUE.

Line 74: IWUE is mentioned but not defined. It is defined later in the text, but it helps the reader to understand of is being described at this point.

―――――――――――――――――

---

## Referee Comment (RC2) · Anonymous Referee #3 · 9 Oct 2016

Currently, global carbon and water cycles as well as the associated carbon-water coupling relationships are receiving more and more attentions in this research field. Particularly under the changing climate, how extreme climate events affect the carbon sequestration of terrestrial forests directly related to the future climate projections. This study aimed to reveal the responses of ecosystem water-use efficiency to summer drought in northern Europe, which will also enrich this hot topic. The work is well organized with clear structure. However, plenty of questions still need to be solved.

Major comments:

1. In the part of "Introduction", the authors used a lot of sentences associated with ecosystem water-use efficiency, such as WUE, EWUE, and IWUE. Moreover, they want to express the potential effects of drought on different WUE expressions. I strongly sug-

gest the authors to explain the differences of various WUE definitions, and the reasons adopted in the present study. Then, the authors used many phrases to describe soil water status. However, it is crucial that how do they define the droughts, especially severe or moderate.

2. In the part of "2.5 Soil Moisture Index (SMI)", the authors used SMI derived from simulated soil moisture and soil parameters in JSBACH to define soil moisture conditions. It may lead to the uncertainty owing to the model performance and the results in this study. Then, why not validate the accuracy of the modeled data. The SMI results are also classified as severe drought, moderate drought, mid-range, moderate wet and very wet. However, only 11-year dataset for Hyytiala and 8-year dataset for Sodankyla, which may be not enough for drought analyses.

3. In the part of "Results and Conclusion", the authors concluded that based on observed data, the ecosystem level water use efficiency (EWUE) showed a decrease only during a severe soil moisture drought at Hyytiälä, whereas the inherent water use efficiency (IWUE) increased when there was a severe soil moisture drought at Hyytiälä and a moderate soil moisture drought at Sodankylä. However, on one hand, this study is based on "no severe soil moisture deficit at Sodankylä during the study period", on the other hand, it seems to be a lack of persuasion. Maybe more data need to be supplemented to enrich the objectives of this study.

4. In the part of "Discussion", I suggest the authors to supplement the uncertainties of SMI, which is used as drought indicator of soil water. In addition, a lot of biotic and abiotic factors controls the differences of GPP, ET, WUE of the two forest sites. So, how do they directly compare the effects of drought on Scots pine forest under different local environmental conditions. Apart from this, I think in northern Europe such as Finland, temperature anomaly may be more sensitive than drought.

Minor comments:

1. P1 L13 & L23 "at daily time scales" should be changed to "at the daily time scale".

2. L15-L18 This long sentence is too tedious. Please reconstruct it.

3. P2 L37 Terrestrial plants assimilate carbon dioxide ($CO_2$)...

4. L59 EWUE is broadly adopted as a surrogate for WUE due to data availability. Why? This expression is abrupt without explanation.

5. L65 It is inappropriate to cite the paper of Reichstein et al. 2002. Ecosystem water-use efficiency of gross carbon uptake decreased during the drought, regardless whether evapotranspiration from eddy covariance or transpiration from sapflow had been used for the calculation.

6. P3L90 Lack of the information on forest ages between the two flux sites.

7. P4L112 "For our analysis, daily values of GPP and ET fluxes were calculated as daily sums of half-hourly values and only good quality gap-filled data were used." It is confusing. How do they guarantee the comparability of different days?

8. P4L131 Crammer mistakes. "The models of Farquhar et al. (1980) is used for photosynthesis of C3 plants."

9. P6L188 "However, process-based ecosystem models can be used to reveal plant physiological processes by separating evaporation and transpiration." Please cite the associated references.

10. P9L311 "The consequence of decreased ET due to soil moisture drought would be increased atmospheric VPD, which in turn accelerates stomatal closure..." Please reconstruct it.

11. Please specify the time scale of data used in the figures. In addition, the data points are too dense. It is hard to extract useful information from the merged figures.

12. In Figure 4, many values of WUE at the daily time scale exceed 10 g C kg-1 H2O, especially for simulated results of EWUE and IWUE. Do you think it is rational or just statistical?

---

## Referee Comment (RC3) · Anonymous Referee #4 · 19 Oct 2016

The article presents an analysis of GPP, ET and water use efficiency metrics for two flux-towers in Finland in search for drought effects or more generally for soil moisture controls on the carbon and water fluxes. The climate and hydrological regime of these sites restrain appreciable effects of water limitations on GPP, ET and water use efficiency to few days during summer 2006 in the southern site of Hyytiälä (Line 210, Line 241-242, Fig.1 and 3) This is probably the most interesting result of the article but constrains quite significantly the scope of the analysis. The data analysis in Fig. 2 is interesting in certain aspects, but overall, the study leaves me quite doubtful about its novelty in the presented conclusions (see comments below). The link between soil moisture and plant physiologically meaningful thresholds is also very weak. A land-surface model, JSBACH, is also used to reproduce the water and carbon fluxes but serious model limitations, a relatively weak rationale for using the model, and a poor

model-data comparison make this part insufficient.

*Major comments*

1) I struggle to identify the novel conclusions of the manuscript (beyond the presentation of the data themselves). The main conclusions are: (i) There are only few days of water limitations in only one of the two analyzed sites despite the 11 and 8 years of analyzed data. Interesting result but it partially hampers the scope of the article. (ii) IWUE and EWUE are identifying two different aspects of ecosystem response, with the first more appropriate to capture changes in surface conductance. This is of course important but it is expected too because one depends explicitly on VPD and the other does not. (ii) Ecosystem models, in this case JSBACH, need to have a very good representation of stomatal functioning and its dependence with VPD or humidity, which is known since quite some time (Ball et al 1987, Leuning 1995) and widely debated in literature (e.g., Monteith 1995; Damour et al 2010) and actually included in most of the models.

2) In my opinion, the use of model simulations in the article lacks a clear rationale. The model is simply used to run the same period of observations and to reproduce the same variables which are observed (only transpiration and EUWEt, IWUEt are additionally analyzed). Therefore, there is not really a benefit or the idea to use the model for specific numerical experiments that go beyond observations. If the scope is confined to test the model performance only, also in this case there is not a direct comparison with data. No scatter plot to evaluate magnitude, seasonality, or other aspects of model performance is shown. Even the behavior with respect to the driving variables (Rsw, Ta, VPD and an index of soil moisture) is shown only qualitatively and not quantitatively, since the simulated and observed variables are never presented in the same plot (Fig. 2, 3, 4).

3) I also have doubts about the choice of the model. From the manuscript description, JSBACH has only few layers of soils, which do not allow a proper representation of

soil moisture vertical dynamics (Line 237-240) and most importantly, there is no representation of vegetation physiology with regards to water stress (thresholds for stomatal closure or plant vital functions), or at least this is not described in the article. Water content limits rather than more physiologically meaningful water potentials are used, which leaves the doubt if the selected thresholds have any meaning for the plant response to drought or not (e.g., Hsiao 1976).

4) The manuscript is generally decently written but there are parts, (e.g., abstract, introduction) which can be written much better (see minor comments below). Also the choice of the presented material is debatable. For instance, Sodankylä is one of the two presented case studies but nothing about Sodankylä is graphically presented in the main manuscript. Data uncertainty issues are discussed but not represented. Different figures share the same information; those can be better re-organized to highlight some of the main conclusion, which are not so evident from the current Figures (e.g., Line 325-326).

*Specific comments*

Page 1. Line 11. I respectfully disagree with this statement; we have a good knowledge and wide body of literature about the carbon and water coupling, from stomatal level to ecosystems (e.g., Katul et al 2012). What it is still problematic is the modeling of the response of vegetation to periods of water stress at different temporal scales (from hourly to multiannual) and at different spatial scale (from a single tree to a region).

Page 1. Line 18-20. This sentence is very badly phrased, what does exactly mean that the decrease in ET is alleviated by increased VPD? If a decrease in ET is observed, this is already implicitly account for changes in VPD. The authors are here referring to the difference between EWUE and IWUE, with the first affected by VPD, why the latter is independent and therefore more indicative of how surface conductance changes. This is, however, not clear from the text.

Page 1. Line 25. What do the authors mean with "deviated groups"? This is explained

only much later in the manuscript. I am not a native speaker but the use of the term "deviated groups" appear, at the very least, awkward to me.

Page 2. Line 42. It is not very clear what the authors mean with "physiological stress" but if they refer to impairment of vital functions and plant mortality, I think that physiological stress may occur much later (at much higher levels of water stress) than reduced carbon uptake.

Page 2. Line 46. I think this sentence could be written much better in English.

Page 2. Line 54-55. This sentence is overly approximate. Ecosystem functioning depends on many more factors than WUE (e.g., nutrient dynamics, species competition and forest demography, just to quote some) and WUE is not simply "closely related" to water and carbon cycles but it is the metric which summarizes how the two cycles are coupled, at least at the flux-level.

Page 2. Line 62-73. I think many of these contrasting results can be simply related to the fact that a water-stress, which is perceived by the plants, occurs or not.

Page 3. Line 96. I wonder if the choice of reporting LAI as "all-sided" rather than as "one-side/projected" as typically done in most of the literature (including for the very same sites, Lindroth et al 2008) is a good choice or not. At the very least, this should be clarified in the text and not only on the Table.

Page 4. Line 113-114. This sentence is not very clear to me. How do you distinguish between filled data of "good-quality" and "bad-quality"? Do you mean that that you discard days with observed low-quality data? Do you mean that you gap fill these data? Do you mean that you discard "half-hourly" periods and you average the others?

Section 2.3. I know that JSBACH is an established model, but the model description is extremely synthetic. I would invite the authors to add a bit more of information. For instance, there is no mention of how the hydrological budget is solved. How do JSBACH deal with transpiration, evaporation from ground, from interception, deep

leakage? How root depth-distribution is considered? How vegetation phenology is considered?

Page 5. Line 146. Can the authors better characterize the spin-up? How long did you run? Which period did you use?

Page 5. Line 149. I would state that they were calculated from "observed data" rather than "model forcing". Are there any differences between the two?

Page 5. Line 165-167. In my opinion, this classification of soil moisture conditions is very arbitrary, since there is no explicit link between the thresholds of SMI and plant physiologically meaningful variables such as "soil water potential" or better "leaf water potential". While overall, it is clear that with decreasing SMI drought stress should increase, there is no reason to support that drought stress should start at SMI of 0.1 or 0.4. I would suggest avoiding such classification and just having a continuous variable SMI.

Page 5. Line 171. I do not fully understand the rationale of using $\theta$sat in place of $\theta$fc. Even leaving a part the problematic concept of $\theta$fc (e.g. Assouline and Or 2014), the two values may be quite different and they are not interchangeable.

Page 5. Line 173-174. Also for Sodankylä, I do not understand why the authors did not use a soil water retention curve to link soil water contents to soil water potentials and therefore define some plant-meaningful threshold rather than arbitrary values.

Page 6. Line 184. It is a measure of the inverse of the surface conductance.

Page 6. Line 208-209. "deviated group" is awkward wording.

Page 7. Line 235-236. As a follow up of my previous comment, SMI<0.2 does not necessarily imply a "drought" from the plant point of view and therefore the lack of deviation in GPP, ET, with respect to normal conditions should be expected as a consequence of the "lack of drought".

Page 8. Line 250-251. How many days are those for which we see such a deviation? Are they continuous in time? This is probably one of the most important information to place in the article, which needs to be emphasized.

Page 8. Line 262-263. This is likely a consequence of the cross-correlation between high SMI and unfavorable meteorological conditions (e.g., cold, cloudy).

Page 9. Line 291-294. What is driving this proportionally larger decrease in GPP than ET? Looking from a leaf-level perspective this is hard to reconcile. What is keep sustaining ET if soil moisture is very low? How much is data uncertainty playing a role in such a response? Is it really just the change in VPD?

Page 11. Line 361-365. I agree that is important to highlight uncertainties of EC data, but how do you tackle this problem in the result presentation? There is not any confidence bound around the data, therefore we cannot really establish if some of data-driven result are very robust or not. At the very least, there should be a more quantitative discussion of the uncertainties.

*References*

Assouline, S., and D. Or (2014), The concept of field capacity revisited: Defining intrinsic static and dynamic criteria for soil internal drainage dynamics, Water Resour. Res., 50, doi:10.1002/2014WR015475.

Ball, J., Woodrow, I., Berry, J., (1987). A model predicting stomatal conductance andits contribution to the control of photosynthesis under differentenvironmental conditions. In: Progress in Photosynthesis Research: Volume 4Proceedings of the VIIth International Congress on Photosynthesis Providence,Springer Netherlands, pp. 221–224

Damour G, Simonneau T, Cochard H, Urban L. (2010). An overview of models of stomatal conductance at the leaf level. Plant Cell Environ 33:1419–1438. doi:10.1111/j.1365-3040.2010.02181.x.

Hsiao TC, Acevedo E, Fereres E, Henderson DW. (1976) Stress metabolism: water

stress, growth and osmotic adjustment. Philos Trans R Soc Lond B Biol Sci, 273:479–500

Katul GG, Oren R, Manzoni S, Higgins C, Parlange MB (2012). Evapotranspiration: a process driving mass transport and energy exchange in the soil-plantatmosphere-climate system. Rev Geophys, 50: RG3002.

Leuning, R., (1995). A critical appraisal of a combined stomatal-photosynthesismodel for C3 plants. Plant. Cell Environ. 18 (4), 339–355.

Lindroth A, Lagergren F, Aurela M, Bjarnadottir B, Christensen T, Dellwik E, Grelle A, Ibrom A, Johansson T, Lankreijer H, Launiainen S, Laurila T, Molder M, Nikinmaa E, Pilegaard K, Sigurdsson BD, Vesala T (2008) Leaf area index is the principal scaling parameter for both gross photosynthesis and ecosystem respiration of Northern deciduous and coniferous forests. Tellus Ser B Chem Phys Meteorol 60:129–142

Monteith J.L. (1995) A reinterpretation of stomatal responses to humidity. Plant, Cell Environment 18, 357–364.

---

## Referee Comment (RC4) · Anonymous Referee #5 · 23 Oct 2016

General

The general topic of the manuscript falls well within the scope of the journal. The manuscript presents an interesting study where the authors used eddy covariance (EC) method and a land surface model (JSBACH) to assess the response of daily water use efficiency (WUE; both ecosystem and intrinsic) to drought in two boreal forest ecosystems in Finland. The manuscript is well-written but may benefit from carful editing. The manuscript contains some interesting results, and I recommend addressing some comments and suggestion raised below to make it clearer.

Specific comments

Was nighttime data included in the analysis? Obviously, there is no assimilation during the night and the EC system may produce or the LSM may simulate negative ET values. This may not be easily detected if daily flux sums, as opposed to sub-daily fluxes, were used but the computed WUE values would be fundamentally flawed and have no meaning. Some of the simulated results still contain negative ET values but no corresponding WUE (IWUE) values are shown for such data in the graphs. I recommend that this issue be addressed in the manuscript.

It is not clear how the authors treated data during rainy days. Rainy days may underestimate WUE as evaporation from the canopy surface would be high during such times. It is vital that the conclusions drawn in this study do not include such low WUE values.

Lines 109–114: Could the authors provide information as what percentage of the data was observed and/or gap-filled for $CO_2$ and $H_2O$ fluxes. The authors stated that only good quality gap-filled data were used in the analysis. It is not clear whether data were discarded at half-hourly or daily time scale; or what percentage of 'good quality gap-filled data' was regarded good enough to be used in the analysis.

Lines 115–118 and lines 147–150: Were these ($T_a$, $R_s$ and VPD) daily averages? If so please specify. Also in all the graphs where these weather variables are used. I would prefer total daily $R_s$ to daily average $R_s$ values though. Do observations and simulations use the same weather variables (I would assume so)? However, I see quite a number of 0 W m-2 $R_s$ observations (at Hyytiälä which is south of Sodankylä!) but not so many in the simulations. These could be because the corresponding simulated y-axis variables were 0 as well. The question is how possible is it to have daily $R_s$ of 0 W m-2 in the summer.

Lines 119–122. Time Domain Reflectometry (not Reflection). Also theta probe is not a technique, it is a sensor. It would be better if you mention the names (models) of the sensors and manufacturers, and readers could find out the techniques should they be interested.

As the soil water content forms the core of the simulation results, it would be helpful if the authors could provide more information on sensor types, how they were installed

(vertically, horizontally or make use of access tube), if they were static or portable, measurement (time) interval, replications at a site, etc. Besides, was the soil water content for each sensor and depth calibrated against standard gravimetric measurements?

Why was the 0 to -5-cm soil water content at Hyytiälä disregarded in the observed vs simulated comparisons?

Lines 161–167: Again, it is not clear why observed and simulated SMI for layer 1 were not used, at least for Hyytiälä. And, how could the observed and simulated SMI be compared when the depth of the third layer is different for the two (at Hyytiälä)?

165–179: Lack of measured soil parameters was the reason behind adopting different soil moisture classification units for the two locations. If there was continuous long-term soil water content data for so many years, it would not have been so difficult to derive these soil parameters (at Sodankylä). Saturated and field capacity soil water content may be estimated in spring following snowmelt, and wilting point from extended dry periods later in summer.

Assuming the soils are similar at the two sites, it seems unlikely that the very wet conditions at Sodankylä would only have a soil water content of 13 to 16

Lines 194–199: Could the authors provide R2 and P values for the graphs? What was the improvement in R2 and P values due to adopting exponential rather than linear relationship between GPP and ET or T?

Lines 205–208: In general, there are more points in zone (b) than in zone (a); and zone (b) includes a larger range of the environmental variables. Both zones share more common than different environmental variables. I wonder if the environmental variables could be used at all to tease out differences in drivers between the two zones (apart from SMI).

Line 222: There are negative simulated ET values (Figs. S1, S2, S5, etc.). Could the authors explain how they treated such data when determining WUE?

Lines 294–297: It would be better if the y-axis of Figs. 4 and S5 were IWUE rather than GPP×VPD. As it is, it is difficult to tell whether IWUE increases or decreases with ET.

Technical comments

Be consistent in using either soil water content or soil moisture throughout.

Line 18 to 20: This is not clear at all. Rewrite please.

Line 58: Change "... the ratio of GPP and ET." to "... the ratio of GPP to ET."

Lines 80 to 83: Does not read well. Consider rewordingĂň–for example to "... the various land ecosystem model simulations highlight the current uncertainty with regard to plant physiology (water use) in response to drought."?

Lines 91 and 92: Consider removing this sentence and start with the actual experimental sites.

Line 93: 'annual temperature' should read 'annual air temperature'. Also in Table 1 (Lines 600–605).

Line 116: Humidity is ambiguous term to use. Be specific which measure of atmospheric humidity was used.

Line 153: Delete the phrase 'to be able'

Line 157: "... ] and $\theta$WILT is the ..."

Line 170: "... (i.e., soil water content ..."

Line 170: "... and $\theta$WILT= ...)"

Line 171–172: Confusing statement. This changes the definition of SMI and probably the boundaries of SMI set-up above. And, if $\theta$FC acts as a proxy for $\theta$SAT, then should not $\theta$FC be used instead of $\theta$SAT?

Line 192 onwards: ET/T is ambiguous. Use ET or T instead.

Line 248: Replace 'slop' with 'slope' in Tables S1 and S2.

Line 300: Simulated EWUE is not presented in Fig. S5.

Lines 304–305: This sentence is not a result. Move it to the Discussion Section.

Lines 313, 320 and 337: Consider changing the word 'disturbance'. Some suggestions: Line 313: 'Moreover, GPP and ET were decoupled and EWUE decreased . . .', Line 320: '. . . there was no deviation in GPP, . . .', and Line 337: 'The simulated daily ET data contained frequent . . .'.

Line 341: '. . . limitations on GPP and ET or T relationships under . . .'.

Line 344: '. . . when soil moisture was under . . .'.

Line 370: Delete 'as a whole'

---

## Referee Comment (RC5) · Anonymous Referee #2 · 29 Oct 2016

**General comments**

In their paper \*Response of water use efficiency to summer drought in boreal Scots pine forests in Finland\* Gao et al. address a timely problem in biogeochemistry, the interaction of the carbon with the water cycles. Knowledge of this relationship is particularly uncertain during periods of water stress, for which the exact physiological mechanisms and their ecological variability are unknown.

Nevertheless, I have the following major comments that I think should be addressed:

1. The authors should discuss why IWUE was chosen as a metric in addition to WUE. A recent study (Zhou et al. 2015) demonstrated that a definition based on a square-root relationship with VPD is superior to the definition of Beer et al. (2009). Notably, the latter is already expected to be dependent on VPD, as stomata react to this variable

and thus the surface conductance changes accordingly.

2. Lines 184-190 address the problem of soil evaporation. While it is true that model predictions of transpiration can be used as proxy variable, this comes at the cost of additional model uncertainties. The cited paper of Beer et al. (2009), which establishes the concept of IWUE, tries to circumvent this problem by excluding days following precipitation events. Most of the excluded days would lie outside dry spells, hence retaining sufficient sample size for these periods. The data presented in the current manuscript could be filtered according to such a criterion; then it would be important to see whether the observed patterns persist or change in magnitude. Generally this approach would be more robust than basing the IWUEt/EWUEt estimates on a model with known deficiencies.

3. I think it is questionable that daily averages were used for the analyses. Especially in light of problems such as dew-fall it would make sense to use day-night-time separated data for the analyses. At least, the absence of day-night-time separation should be mentioned in the text.

4. Regarding the effect of atmospheric humidity the text states that "Our results indicate that the combined effects of soil moisture and atmospheric drought on stomatal conductance have to be taken into account." (ll.351-353) I think the current version of the text doesn't fully establish the interaction and correlation between the atmospheric and subsurface stress factors. The observed effects by themselves are not unexpected, as the model in its current form simply lacks the stomatal response to atmospheric humidity.

5. The paragraph in ll.325-329 is confusing. First, the statement that "This means that the intrinsic water use efficiency at the ecosystem level is enhanced during soil moisture drought." is merely restating the increase already mentioned in the preceding sentence. Further, wouldn't one expect that a better adaptation to drought leads to elevated IWUE, rather than interpreting a constant IWUE as the sign for this adaptation?

Aside from that, it could be worth commenting on whether differences in adapation between the southern and northern site would be expected *a priori*, e.g. due to average recurrence times of droughts at these locations.

6. Generally, the inclusion and evaluation of JSBACH simulations would profit from a more targeted motivation. What is the predicted behavior? What is already known? In what way could the presented analysis contribute to an improvement of the model? In addition to that, the formulation used for the effect of soil moisture on stomatal conductance should be stated.

**Specific & minor comments**

- The authors state correctly that "there may be systematic errors source from imperfect spectral corrections and gap-filling procedures or calibration problems" (ll.364-365). This would make it all the more important to report which exact criteria were used to exclude observations with insufficient data quality.

- l.109: Which partitioning method was used? Should be mentioned and cited in the text.

- ll.379-381: "Also, in the relationships between ET/T and VPD at the two sites, both observed and simulated ET/T showed a small decrease under moderate soil moisture drought, compared to days with higher soil moisture conditions." It is not clear from this sentence, whether this relates to the sensitivity of ET to VPD or ET itself.

- The limitations of the EC method mentioned in ll.361-365 are true by itself, however in the current text they appear very unconnected to any discrepancies or problems in the analyses. If specific problems of the method, such as the energy-balance-closure-gap can be made responsible for particular deviations, that should be connected in the discussion. Else, the part can be shortened and moved to the methods section.

**Technical corrections**

- "In addition, there may be systematic errors source from imperfect spectral corrections

and gap-filling procedures or calibration problems" (ll.364-365) should be changed to "In addition, imperfect spectral corrections and gap-filling procedures as well as calibration problems may be sources of systematic errors."

**Citations**

Zhou et al. (2015). Daily underlying water use efficiency for AmeriFlux sites. *Journal of Geophysical Research: Biogeosciences*. DOI: 10.1002/2015JG002947.
* * *

---

## Author Comment (AC1) · 18 Apr 2017

General comments:

This paper deals with an important issue relating the water and carbon cycles on the ecosystem level. The authors used two long-term datasets and derived interesting results about water use efficiency.

The introduction is good, explaining well the problem and citing many recent references. The methodology describes well the sites characteristics and the methods used.

Finally, the text is well written and the quality of figures is good.

Specific comments:
Abstract Lines 16-17: It would be better to have a sentence or two explaining how EWUE and IWUE were defined, since different definitions might be used such as NEE/ET for EWUE.

Authors response (AR): The explanations of how EWUE and IWUE were defined have been added.

Line 74: IWUE is mentioned but not defined. It is defined later in the text, but it helps the reader to understand of is being described at this point.

AR: The definition of IWUE has been given at this point in the text.

Please also note the supplement to this comment:
http://www.biogeosciences-discuss.net/bg-2016-198/bg-2016-198-AC1-supplement.pdf

**Supplement:**

*We deeply appreciate all the reviewers for their constructive comments in improving the scientific quality of this manuscript. Our point-by-point response to all the reviewers' comments are listed below, and corresponding modifications are also made for the manuscript. We hope our reply will satisfy the expectations from reviewers.*

**Response to Anonymous Referee #1**

**General comments:**

This paper deals with an important issue relating the water and carbon cycles on the ecosystem level. The authors used two long-term datasets and derived interesting results about water use efficiency.

The introduction is good, explaining well the problem and citing many recent references. The methodology describes well the sites characteristics and the methods used.

Finally, the text is well written and the quality of figures is good.

**Specific comments:**

Abstract Lines 16-17: It would be better to have a sentence or two explaining how EWUE and IWUE were defined, since different definitions might be used such as NEE/ET for EWUE.

*Authors response (AR): The explanations of how EWUE and IWUE were defined have been added.*

Line 74: IWUE is mentioned but not defined. It is defined later in the text, but it helps the reader to understand of is being described at this point.

*AR: The definition of IWUE has been given at this point in the text.*

**Response to Anonymous Referee #2**

**General comments :**

In their paper *Response of water use efficiency to summer drought in boreal Scots pine forests in Finland* Gao et al. address a timely problem in biogeochemistry, the interaction of the carbon with the water cycles. Knowledge of this relationship is particularly uncertain during periods of water stress, for which the exact physiological mechanisms and their ecological variability are unknown.

Nevertheless, I have the following major comments that I think should be addressed:

1. The authors should discuss why IWUE was chosen as a metric in addition to WUE. A recent study (Zhou et al. 2015) demonstrated that a definition based on a square-root relationship with VPD is superior to the definition of Beer et al. (2009). Notably, the latter is already expected to be dependent on VPD, as stomata react to this variable and thus the surface conductance changes accordingly.

*AR: In the revised paper, we added the underlying water use efficiency (uWUE) that introduced by Zhou et al. (2014) for a comparison with WUE and IWUE for studying their performances during the summer drought. IWUE is defined as WUE multiplied with mean daylight vapor pressure deficit (VPD), and it has been found to increase during short-term moderate drought. In the formulation of IWUE, ET/VPD is a hydrological measure of the surface conductance at the ecosystem level (Beer et al., 2009). The uWUE is proposed based on IWUE and a simple stomatal model of Lloyd and Farquhar (1994). Different to IWUE which is still affected by the nonlinear effect of VPD, the uWUE has been found to represent the best linear relationship among GPP, ET and VPD at the half-hourly time scale by Zhou et al. (2014). Later on, the appropriateness of uWUE at daily time scale has been demonstrated (Zhou et al., 2015). However, we were not clear how uWUE behaves during drought period. In our study, we found that uWUE doesnot show a change during the short-term summer drought at our site. As uWUE is more independent of VPD, it is considered that uWUE is more suitable as a plant functioning metric to evaluate the impact of global change on plant functioning at ecosystem level in the long term. Those contents have been introduced and discussed in the revised manuscript.*

2. Lines 184-190 address the problem of soil evaporation. While it is true that model predictions of transpiration can be used as proxy variable, this comes at the cost of additional model uncertainties. The cited paper of Beer et al. (2009), which establishes the concept of IWUE, tries to circumvent this problem by excluding days following precipitation events. Most of the excluded days would lie outside dry spells, hence retaining sufficient sample size for these periods. The data presented in the current manuscript could be filtered according to such a criterion; then it would be important to see whether the observed patterns persist or change in magnitude. Generally this approach would be more robust than basing the IWUEt/EWUEt estimates on a model with known deficiencies.

*AR: Yes, we filtered our data to exclude the rainy days and the certain amount of dry days after the rainy days in the revised manuscript. As suspected by the reviewer, sufficient sample size for the drought period retain after the data selection. We could observe a more significant pattern of the impacts on GPP and ET from the soil moisture drought.*

3. I think it is questionable that daily averages were used for the analyses. Especially in light of problems such as dew-fall it would make sense to use day-night-time separated data for the analyses. At least, the absence of day-night-time separation should be mentioned in the text.

*AR: The data has now been reprocessed for the analysis, and only daytime data without precipitation influence were selected. The data selection process is described in section 2.2. In the revised manuscript, only half-hourly data with shortwave radiation (Rs) larger than 100 $W/m^2$ were selected for the aim to select the effective time for plant photosynthesis. The rainy days and certain amount of dry days after the rainy days were also excluded. By doing this, data with negative GPP and ET were excluded.*

4. Regarding the effect of atmospheric humidity the text states that "Our results indicate that the combined effects of soil moisture and atmospheric drought on stomatal conductance have to be taken into account." (ll.351-353) I think the current version of the text doesn't fully establish the interaction and correlation between the atmospheric and subsurface stress factors. The observed effects by themselves are not unexpected, as the model in its current form simply lacks the stomatal response to atmospheric humidity.

*AR: Yes, the model in its current form lacks the stomatal response to atmospheric humidity. In global ecosystem models, simple representations of stomatal regulation have often been applied to reduce*

*computing costs. Because VPD and soil moisture are to certain degree correlated, inclusion of one of the either has often shown to be enough to account for drought effects. In the revised manuscript, the formulations of the default stomatal conductance model in JSBACH has been added. It can be found that the soil moisture condition is the only limiting factor in the default stomatal conductance model in JSBACH. Knauer et al. (2015) tested a few stomatal conductance models in the JSBACH model under non-limited soil moisture conditions, and the results showed that Ball-Berry model (Ball et al., 1987) to be best in its response to atmospheric drought. However, the performance of the default stomatal conductance model under limited soil moisture conditions has not been tested before this study. Our results showed that the model can successfully capture the turning point of GPP and ET when the SMI decreased to be lower than 0.2. However, the decreases of GPP and ET are not as strong as in the observations. Thus, our results indicate that the combined effects of soil moisture and atmospheric drought on stomatal conductance have to be both taken into account. Even though no such a correlation between the stress factors was established in our study, it was demonstrated that at certain point the correlation between GPP or ET and soil moisture or VPD vanishes. So it is insufficient to use only soil moisture or VPD to describe drought stress.*

5. The paragraph in ll.325-329 is confusing. First, the statement that "This means that the intrinsic water use efficiency at the ecosystem level is enhanced during soil moisture drought." is merely restating the increase already mentioned in the preceding sentence. Further, wouldn't one expect that a better adaptation to drought leads to elevated IWUE, rather than interpreting a constant IWUE as the sign for this adaptation? Aside from that, it could be worth commenting on whether differences in adaption between the southern and northern site would be expected *a priori*, e.g. due to average recurrence times of droughts at these locations.

*AR: Yes, we agree with the reviewer the sentence is redundant and we have revised this paragraph. However, we do not agree with reviewer on the assumption that elevated IWUE would necessarily be a sign of adaptation. The trees might also be opportunistic in their strategies. The different behaviors of IWUEs imply different strategies in the south than in the north.*

6. Generally, the inclusion and evaluation of JSBACH simulations would profit from a more targeted motivation. What is the predicted behavior? What is already known? In what way could the presented analysis contribute to an improvement of the model? In addition to that, the formulation used for the effect of soil moisture on stomatal conductance should be stated.

*AR: For our reply to the first part of this comment, please refer to our answer for the comment 4 above. Additionally, we have added a section 2.3.1 in the revised manuscript to describe the stomatal conductance model in JSBACH.*

**Specific & minor comments:**

- The authors state correctly that "there may be systematic errors source from imperfect spectral corrections and gap-filling procedures or calibration problems" (ll.364-365). This would make it all the more important to report which exact criteria were used to exclude observations with insufficient data quality.

- l.109: Which partitioning method was used? Should be mentioned and cited in the text.

*AR: For Hyytiälä, EC fluxes were calculated using standard methods as described in Mammarella et al (2016). Data quality of 30 min values of NEE and latent heat flux (LE) was ensured excluding records*

*with low turbulent mixing (friction velocity below 0.25 m/s) as described in Markkanen et al. (2001), Mammarella et al (2007) and Ilvesniemi et al. (2010). The NEE was partitioned into Re and GPP according to Kolari et al. (2009). Shortly, Re was modelled using an exponential equation with temperature at a depth of 2 cm in the soil organic layer as the explanatory factor. The value of GPP was then directly derived as residual from the measured NEE. When NEE was missing, GPP was estimated according to Eq.7 in Kolari et al. (2009). LE was gap-filled using a linear regression against net radiation in a moving window of 5 days. Then ET is converted from LE. We have added these details and the missing references in the revised manuscript.*

*Kolari, P., Kulmala, L., Pumpanen, J., Launiainen, S., Ilvesniemi, H., Hari, P., and Nikinmaa, E.: CO2 exchange and component CO2 fluxes of a boreal Scots pine forest, Boreal Environment Research, 14, 761- 783, 2009.*

*Mammarella, I., Peltola, O., Nordbo, A., Järvi, L., and Rannik, Ü.: Quantifying the uncertainty of eddy covariance fluxes due to the use of different software packages and combinations of processing steps in two contrasting ecosystems, Atmos. Meas. Tech., 9, 4915-4933, doi:10.5194/amt-9-4915-2016, 2016.*

*Markkanen, T., Rannik, U., Keronen, P., Suni, T., and Vesala, T.: Eddy covariance fluxes over a boreal Scots pine forest, Boreal Environment Research, 6, 65-78, 2001.*

*Ilvesniemi, H., Pumpanen, J., Duursma, R., Hari, P., Keronen, P., Kolari, P., Kulmala, M., Mammarella, I., Nikinmaa, E., Rannik, U., Pohja, T., Siivola, E., and Vesala, T.: Water balance of a boreal Scots pine forest, Boreal Environment Research, 15, 375-396. 2010.*

*Mammarella, I., Kolari, P., Rinne, J., Keronen, P., Pumpanen, J. and Vesala, T.: Determining the contribution of vertical advection to the net ecosystem exchange at Hyytiälä forest, Finland, Tellus B, 59, 900-909, doi:10.1111/j.1600-0889.2007.00306.x, 2007.*

- ll.379-381: "Also, in the relationships between ET/T and VPD at the two sites, both observed and simulated ET/T showed a small decrease under moderate soil moisture drought, compared to days with higher soil moisture conditions." It is not clear from this sentence, whether this relates to the sensitivity of ET to VPD or ET itself.

*AR: We agree with the reviewer that the sentence is not clear enough. It was referred to the sensitivity of ET to VPD. We have revised this part also because the data was reprocessed for the revised manuscript.*

- The limitations of the EC method mentioned in ll.361-365 are true by itself, however in the current text they appear very unconnected to any discrepancies or problems in the analyses. If specific problems of the method, such as the energy-balance-closure- gap can be made responsible for particular deviations, that should be connected in the discussion. Else, the part can be shortened and moved to the methods section.

*AR: Yes, we will move this part to the methods section in the revised manuscript.*

**Technical corrections :**

- "In addition, there may be systematic errors source from imperfect spectral corrections and gap-filling procedures or calibration problems" (ll.364-365) should be changed to "In addition, imperfect spectral corrections and gap-filling procedures as well as calibration problems may be sources of systematic errors."

*AR*: *We did the change as reviewer suggested.*

**Citations**
Zhou et al. (2015). Daily underlying water use efficiency for AmeriFlux sites. *Journal of Geophysical Research: Biogeosciences*. DOI: 10.1002/2015JG002947.

**Response to Anonymous Referee #3**

Currently, global carbon and water cycles as well as the associated carbon-water coupling relationships are receiving more and more attentions in this research field. Particularly under the changing climate, how extreme climate events affect the carbon sequestration of terrestrial forests directly related to the future climate projections. This study aimed to reveal the responses of ecosystem water-use efficiency to summer drought in northern Europe, which will also enrich this hot topic. The work is well organized with clear structure. However, plenty of questions still need to be solved.

**Major comments:**

1. In the part of "Introduction", the authors used a lot of sentences associated with ecosystem water-use efficiency, such as WUE, EWUE, and IWUE. Moreover, they want to express the potential effects of drought on different WUE expressions. I strongly suggest the authors to explain the differences of various WUE definitions, and the reasons adopted in the present study. Then, the authors used many phrases to describe soil water status. However, it is crucial that how do they define the droughts, especially severe or moderate.

*AR: The definitions and background of those WUE metrics have been given in the revised introduction. In the revised manuscript, we use the soil moisture index (SMI) to describe the soil moisture status rather than the different droughts at the study site. Thus, in the revised manuscript, the soil moisture conditions were classified into five SMI groups with an interval of 0.2: very dry: $0 \leq SMI < 0.2$, moderate dry: $0.2 \leq SMI < 0.4$, mid-range: $0.4 \leq SMI < 0.6$, moderate wet: $0.6 \leq SMI < 0.8$, very wet: $0.8 \leq SMI < 1$.*

2. In the part of "2.5 Soil Moisture Index (SMI)", the authors used SMI derived from simulated soil moisture and soil parameters in JSBACH to define soil moisture conditions. It may lead to the uncertainty owing to the model performance and the results in this study. Then, why not validate the accuracy of the modeled data. The SMI results are also classified as severe drought, moderate drought, mid-range, moderate wet and very wet. However, only 11-year dataset for Hyytiala and 8-year dataset for Sodankyla, which may be not enough for drought analyses.

*AR: In section 3.1 of the revised manuscript, the simulated SMI that is calculated with simulated soil moisture and soil parameters in JSBACH has been compared with the observed SMI based on observed soil moisture and measured soil parameters. The simulated SMI agreed well with in situ observed SMI over the 11-year study period, with a correlation coefficient (0.625) and a root-mean-square error (RMSE) of 0.225. Moreover, a very good time correlation (0.965) between simulated and observed SMIs were found for year 2006, despite the simulated SMI is systematically lower than the observed*

*SMI (RMSE = 0.12). As we have answered to the above major comment from reviewer #3, the five SMI groups were renamed to represent different soil moisture conditions rather than drought conditions. Unfortunately, we have used the entire data period of the measured soil moisture at the sites. However, Muukkonen et al. (2015) showed that the summer drought in 2006 has caused severe forest damages in southern Finland. Using SMI calculated from regional soil moisture simulations over the past 30 years (1981-2010), such extreme drought affecting forest health has been illustrated to be rare in Finland, and the summer drought in 2006 was the most severe one in the 30-year study period (Gao et al., 2016). In the revised manuscript, we had more detailed analysis of the severe summer drought in 2006 at Hyytiälä, and the Sodankylä site was not included in the paper anymore due to no such severe soil moisture drought in the study period.*

3. In the part of "Results and Conclusion", the authors concluded that based on observed data, the ecosystem level water use efficiency (EWUE) showed a decrease only during a severe soil moisture drought at Hyytiälä, whereas the inherent water use efficiency (IWUE) increased when there was a severe soil moisture drought at Hyytiälä and a moderate soil moisture drought at Sodankylä. However, on one hand, this study is based on "no severe soil moisture deficit at Sodankylä during the study period", on the other hand, it seems to be a lack of persuasion. Maybe more data need to be supplemented to enrich the objectives of this study.

*AR: Yes, we agree with the reviewer that sentence was a bit misleading because there was no such severe drought at Sodankylä during the study period. Please also see the answer to the previous comment.*

4. In the part of "Discussion", I suggest the authors to supplement the uncertainties of SMI, which is used as drought indicator of soil water. In addition, a lot of biotic and abiotic factors controls the differences of GPP, ET, WUE of the two forest sites. So, how do they directly compare the effects of drought on Scots pine forest under different local environmental conditions. Apart from this, I think in northern Europe such as Finland, temperature anomaly may be more sensitive than drought.

*AR: As answered to the comment 1, we realized that it is inappropriate to use fixed SMI groups to describe different drought conditions. However, SMI that ranges from 0 to 1 describes the status of soil moisture available to plants. How to divide the groups of SMI is a matter of choice. In the revised manuscript, we grouped SMI with an interval of 0.2 to reflect different soil moisture status: very dry: 0 ≤ SMI < 0.2, moderate dry: 0.2 ≤ SMI < 0.4, mid-range: 0.4 ≤ SMI < 0.6, moderate wet: 0.6 ≤ SMI < 0.8, very wet: 0.8 ≤ SMI <1.*

*We also agree with the reviewer's opinion that it is a bit uncertain to draw the conclusion about the comparison between the southern site and northern site (scots pine has weaker response to drought in the southern site than in the northern site) by analysing current available data, especially there is no severe soil moisture drought in the studied period in the northern site. Therefore, in the revised manuscript, we focused on the drought event and its impact on plant functioning in the southern site.*

**Minor comments:**

1. P1 L13 & L23 "at daily time scales" should be changed to "at the daily time scale".

*AR: We did the change as reviewer suggested.*

2. L15-L18 This long sentence is too tedious. Please reconstruct it.

*AR: The sentence has been reformulated according to the revised results.*

3. P2 L37 Terrestrial plants assimilate carbon dioxide (CO2)...

*AR: We did the change according to this comment in revised manuscript.*

4. L59 EWUE is broadly adopted as a surrogate for WUE due to data availability. Why? This expression is abrupt without explanation.

*AR: The sentence has been reformulated as: "EWUE is broadly adopted as a surrogate for the leaf level WUE because more data are available at the ecosystem level than at the leaf level.".*

5. L65 It is inappropriate to cite the paper of Reichstein et al. 2002. Ecosystem water-use efficiency of gross carbon uptake decreased during the drought, regardless whether evapotranspiration from eddy covariance or transpiration from sapflow had been used for the calculation.

*AR: Indeed, the reference was misused. we deleted this reference in the revised manuscript.*

6. P3L90 Lack of the information on forest ages between the two flux sites.

*AR: We have given the forest age information in the revised manuscript.*

7. P4L112 "For our analysis, daily values of GPP and ET fluxes were calculated as daily sums of half-hourly values and only good quality gap-filled data were used." It is confusing. How do they guarantee the comparability of different days?

*AR: The original sentence was confusing and not total correct. It has been deleted in the revised manuscript. Actually, the bad quality data was also gap-filled, and the gap-filled data were used. The gap filling method for GPP and ET was introduced in the manuscript. Thus there is no problem with the comparability of different days.*

8. P4L131 Grammar mistakes. "The models of Farquhar et al. (1980) is used for photosynthesis of C3 plants."

*AR: We have corrected the sentence to be " The models of Farquhar et al. (1980) and Collatz et al. (1992) is used for photosynthesis of C3 and C4 plants, respectively."*

9. P6L188 "However, process-based ecosystem models can be used to reveal plant physiological processes by separating evaporation and transpiration." Please cite the associated references.

*AR: This sentence should be reworded as: "However, process-based ecosystem models do resolve evaporation and transpiration, which together composing evapotranspiration". We do not think a reference is needed after reformulation.*

10. P9L311 "The consequence of decreased ET due to soil moisture drought would be increased atmospheric VPD, which in turn accelerates stomatal closure..." Please reconstruct it.

*AR: We have reworded the sentence as "The decrease of ET due to soil moisture drought could lead to*

*increased atmospheric VPD, which in turn intensifies stomatal closure ...”*

11. Please specify the time scale of data used in the figures. In addition, the data points are too dense. It is hard to extract useful information from the merged figures.

*AR: The time scale of data (the averaging period and the study period) has been added in the figures. As the data has been reprocessed in the revised manuscript to exclude rainy days and certain amount of dry days after the rainy days, the amount of data shown on the figures has been reduced. The figures have been reselected and composed in the revised manuscript. We think it should be OK to extract useful information from figures in the revised manuscript.*

12. In Figure 4, many values of WUE at the daily time scale exceed 10 g C kg-1 H2O, especially for simulated results of EWUE and IWUE. Do you think it is rational or just statistical?

*AR: It was quite statistical due to the small ET values, which are mainly impacted by rainy days and days after rainy days with high atmospheric humidity, as well as night time condensation effect. In the revised manuscript, the data has been reprocessed to exclude precipitation and night time impacts on GPP and ET. Thus, there are only three days with EWUE values larger than 10 g C $kg^{-1}$ $H_2O$ for the simulated results in the updated figure, and no EWUE values are larger than 10 g C $kg^{-1}$ $H_2O$ for the observed results.*

**Response to Anonymous Referee #4**

The article presents an analysis of GPP, ET and water use efficiency metrics for two flux-towers in Finland in search for drought effects or more generally for soil moisture controls on the carbon and water fluxes. The climate and hydrological regime of these sites restrain appreciable effects of water limitations on GPP, ET and water use efficiency to few days during summer 2006 in the southern site of Hyytiälä (Line 210, Line 241-242, Fig.1 and 3) This is probably the most interesting result of the article but constrains quite significantly the scope of the analysis. The data analysis in Fig. 2 is interesting in certain aspects, but overall, the study leaves me quite doubtful about its novelty in the presented conclusions (see comments below). The link between soil moisture and plant physiologically meaningful thresholds is also very weak. A land surface model, JSBACH, is also used to reproduce the water and carbon fluxes but serious model limitations, a relatively weak rationale for using the model, and a poor model-data comparison make this part insufficient.

**Major comments:**

1) I struggle to identify the novel conclusions of the manuscript (beyond the presentation of the data themselves). The main conclusions are: (i) There are only few days of water limitations in only one of the two analyzed sites despite the 11 and 8 years of analyzed data. Interesting result but it partially hampers the scope of the article. (ii) IWUE and EWUE are identifying two different aspects of ecosystem response, with the first more appropriate to capture changes in surface conductance. This is of course important but it is expected too because one depends explicitly on VPD and the other does not. (ii) Ecosystem models, in this case JSBACH, need to have a very good representation of stomatal functioning and its dependence with VPD or humidity, which is known since quite some time (Ball et al 1987, Leuning 1995) and widely debated in literature (e.g., Monteith 1995; Damour et al 2010) and actually included in most of the models.

*AR: (i) We have revised the manuscript to only focus on the severe drought in 2006 at Hyytiälä site,*

*because no severe drought took place during the multi-year study period at the northern Sodankylä site. The summer drought in 2006 caused severe forest damages in southern Finland (Muukkonen et al., 2015). Using SMI calculated from regional soil moisture simulations over the past 30 years (1981-2010), such extreme drought affecting forest health has been illustrated to be rare and the summer drought in 2006 in southern Finland was the most severe one in the 30-year study period (Gao et al., 2016). Therefore, it is valuable to study the severe drought in 2006 and its impact on plant functioning.*

*(ii) We aimed to compare the behavior of different water use efficiency metrics under the soil moisture drought. From the literature presented in introduction, we have understood that there is no clear conclusion of the impact of drought on EWUE (increase or decrease), and therefore it is valuable to study the performance of EWUE in drought conditions, as it is widely used metric describing ecosystem level responses. IWUE is a quantity defined as EWUE multiplied with mean daylight vapor pressure deficit and has been found to increase during short-term moderate drought (Beer et al., 2009). In the revised manuscript, we also added uWUE which is developed based on IWUE and a simple stomatal model, to more explicitly assess the role of the stomatal conductance, and to see how uWUE behaves in comparison to the two other metrics.*

*(iii) We agree that it is ideal that ecosystem models can have a very good representation of stomatal functioning and its dependence on VPD. However, in global ecosystem models, simple representations of stomatal regulation have often been applied to reduce computing costs. Because VPD and soil moisture are to certain degree correlated, inclusion of one of the either has often shown to be enough to account for drought effects. In the revised manuscript, the formulations of the default stomatal conductance model in JSBACH has been added. It can be found that the soil moisture condition is the only limiting factor in the default stomatal conductance model in JSBACH. Knauer et al. (2015) tested a few stomatal conductance models in the JSBACH model, and the results showed that Ball-Berry model (Ball et al., 1987) to be best in its response to atmospheric drought under non-limited soil moisture conditions. However, the performance of the default stomatal conductance model under limited soil moisture conditions has not been tested before this study. Our results indicate that the combined effects of soil moisture and atmospheric drought on stomatal conductance have to be both taken into account.*

2) In my opinion, the use of model simulations in the article lacks a clear rationale. The model is simply used to run the same period of observations and to reproduce the same variables which are observed (only transpiration and EWUEt, IWUEt are additionally analyzed). Therefore, there is not really a benefit or the idea to use the model for specific numerical experiments that go beyond observations. If the scope is confined to test the model performance only, also in this case there is not a direct comparison with data. No scatter plot to evaluate magnitude, seasonality, or other aspects of model performance is shown. Even the behavior with respect to the driving variables (Rsw, Ta, VPD and an index of soil moisture) is shown only qualitatively and not quantitatively, since the simulated and observed variables are never presented in the same plot (Fig. 2, 3, 4).

*AR: The purpose of our study are twofold: one is to find how drought influences plant functioning using observational data; the other one is to see if the model can represent the drought and its impact on plant functioning in line with the observational data. In the revised manuscript (section 3.1), the observed SMI and simulated SMI were compared during the study period and drought year. We agree that the GPP and ET changes with respect to the driving variables show how the modelled GPP and ET perform qualitatively, but this (i.e. how modelled GPP and ET changes due to the drought) is what we are mostly interested in. We also think showing how the GPP and ET values differ under different environmental conditions, provide even more useful information for the model improvements than what*

*we can get from correlations by plotting data in the same figure.*

3) I also have doubts about the choice of the model. From the manuscript description, JSBACH has only few layers of soils, which do not allow a proper representation of soil moisture vertical dynamics (Line 237-240) and most importantly, there is no representation of vegetation physiology with regards to water stress (thresholds for stomatal closure or plant vital functions), or at least this is not described in the article. Water content limits rather than more physiologically meaningful water potentials are used, which leaves the doubt if the selected thresholds have any meaning for the plant response to drought or not (e.g., Hsiao 1976).

*AR: Most global land surface models have few layers of soil due to the limitations in computational costs. Nevertheless, drawn from Gao et al. (2016), we can conclude that regionally the model provides relatively robust estimate of SMI. In the revised manuscript section 3.1, good correlation coefficients were found between the simulated SMI and observed SMI over the study period and especially in the drought year. Moreover, there is parameterization of response of vegetation physiology on drought. We have added the formulations of stomatal conductance model in JSBACH as section 2.3.1. Furthermore, the thresholds are evenly set in terms of SMI. SMI is an indicator of soil moisture drought but not the physiological drought. We work more in the meteorological terms than in the plant physiological terms. The deviating responses among different SMI groups as functions of meteorological drivers reflect the response of vegetation. We agree that the original names for the SMI groups easily led to confusion about soil moisture drought or plant physiology drought. Thus, we renamed the SMI groups to describe the soil moisture conditions as very dry, moderate dry, mid-range, moderate wet and very wet.*

4) The manuscript is generally decently written but there are parts, (e.g., abstract, introduction) which can be written much better (see minor comments below). Also the choice of the presented material is debatable. For instance, Sodankylä is one of the two presented case studies but nothing about Sodankylä is graphically presented in the main manuscript. Data uncertainty issues are discussed but not represented. Different figures share the same information; those can be better re-organized to highlight some of the main conclusion, which are not so evident from the current Figures (e.g., Line 325-326).

*AR: We have tried to solve those minor comments below and rewrite parts of the paper. We revised the paper to focus on the severe drought at Hyytiälä site. The Sodankylä site is not included in the analysis anymore as there is no severe drought happened at Sodankylä in the study period. Figures were reselected and recomposed in the revised manuscript.*

**Specific comments:**

Page 1. Line 11. I respectfully disagree with this statement; we have a good knowledge and wide body of literature about the carbon and water coupling, from stomatal level to ecosystems (e.g., Katul et al 2012). What it is still problematic is the modeling of the response of vegetation to periods of water stress at different temporal scales (from hourly to multiannual) and at different spatial scale (from a single tree to a region).

*AR: We tried to simplify this introductory sentence to be: "The influence of drought on plant functioning has received considerable attention in recent years, although our understanding of the response of carbon and water coupling to drought in terrestrial ecosystems still needs to be improved.".*

Page 1. Line 18-20. This sentence is very badly phrased, what does exactly mean that the decrease in ET is alleviated by increased VPD? If a decrease in ET is observed, this is already implicitly account for changes in VPD. The authors are here referring to the difference between EWUE and IWUE, with the first affected by VPD, why the latter is independent and therefore more indicative of how surface conductance changes. This is, however, not clear from the text.

*AR: We wanted to mean that the decrease in stomatal conductance can lead to decreased Transpiration, and low soil moisture can lead to decreased Evaporation. However, as the VPD also increases during the soil moisture drought, the increased VPD could stimulates ET to a certain degree. In general, the ET still decreased during soil moisture drought. We have reformulated this part in the revised manuscript.*

Page 1. Line 25. What do the authors mean with "deviated groups"? This is explained only much later in the manuscript. I am not a native speaker but the use of the term "deviated groups" appear, at the very least, awkward to me.

*AR: The "deviated groups" there referred to the group of data under the severe soil moisture drought. We have deleted this term in the revised manuscript.*

Page 2. Line 42. It is not very clear what the authors mean with "physiological stress" but if they refer to impairment of vital functions and plant mortality, I think that physiological stress may occur much later (at much higher levels of water stress) than reduced carbon uptake.

*AR: Yes, we agree the physiological stress may occur at higher levels of water stress when plant starves as a result of continued metabolic demand for carbohydrates. Therefore, we have revised the sentence as: "... which in turn leads to less carbon uptake (diffusion of $CO_2$ into the leaf) and maybe also subsequent physiological stress …".*

Page 2. Line 46. I think this sentence could be written much better in English.

*AR: We have reformulated the sentence as: "Even though the occurrence of drought is low in northern Europe, the summer of 2006 in Finland has been found to be extremely dry and 24.4 % of the 603 forest health observation sites over entire Finland showed drought damage symptoms in visual examination, in comparison to 2–4 % damaged sites in a normal year (Muukkonen et al. 2015)."*

Page 2. Line 54-55. This sentence is overly approximate. Ecosystem functioning depends on many more factors than WUE (e.g., nutrient dynamics, species competition and forest demography, just to quote some) and WUE is not simply "closely related" to water and carbon cycles but it is the metric which summarizes how the two cycles are coupled, at least at the flux-level.

*AR: We agree with the reviewer. We have reformulated the sentence as: "WUE can be used to study ecosystem functioning which is closely related to the global cycles of water, energy and carbon."*

Page 2. Line 62-73. I think many of these contrasting results can be simply related to the fact that a water-stress, which is perceived by the plants, occurs or not.

*AR: We do not fully agree with the reviewer about this comment. We think the plants do perceive water stress if such conditions have been reported in those studies. Whether the plants react to water stress and how they react to it, depends on the severity of drought, and also other factors such as species (leaf*

*properties, canopy architecture, rooting depth etc) and plants' adaptation to the local climate.*

Page 3. Line 96. I wonder if the choice of reporting LAI as "all-sided" rather than as "one-side/projected" as typically done in most of the literature (including for the very same sites, Lindroth et al 2008) is a good choice or not. At the very least, this should be clarified in the text and not only on the Table.

*AR: We have clarified this in the text.*

Page 4. Line 113-114. This sentence is not very clear to me. How do you distinguish between filled data of "good-quality" and "bad-quality"? Do you mean that that you discard days with observed low-quality data? Do you mean that you gap fill these data? Do you mean that you discard "half-hourly" periods and you average the others?

*AR: We agree the original sentence was confusing and not totally correct. It has been deleted in the revised manuscript. Actually, the bad quality data was gap-filled, and the gap-filled data were used for averages. The gap filling method for GPP and ET was introduced in the manuscript.*

Section 2.3. I know that JSBACH is an established model, but the model description is extremely synthetic. I would invite the authors to add a bit more of information. For instance, there is no mention of how the hydrological budget is solved. How do JSBACH deal with transpiration, evaporation from ground, from interception, deep leakage? How root depth-distribution is considered? How vegetation phenology is considered?

*AR: We have added the descriptions of the stomatal conductance model in JSBACH, which is the most relevant part of the model to this work. We have no enough space in the manuscript to introduce details about soil hydrology and plant phenology of JSBACH model, however, please refer to the literatures cited in our manuscript. The five layer soil hydrology scheme has been introduced in Hagemann and Stacke (2015), and the plant phenology has been described in Böttcher et al. (2016).*

*Böttcher K., Markkanen, T., Thum, T., Aalto, T., Aurela, M., Reick, C. H., Kolari, P., Arslan, A. N., and Pulliainen. J.: Evaluating biosphere model estimates of the start of the vegetation active season in boreal forests by satellite observations, 8, 580, doi:10.3390/rs8070580, Remote Sens., 2016.*

*Hagemann, S. and Stacke, T.: Impact of the soil hydrology scheme on simulated soil moisture memory, Climate Dynamics, 44, 1731-1750, 2015.*

Page 5. Line 146. Can the authors better characterize the spin-up? How long did you run? Which period did you use?

*AR: Prior to the actual simulation, a 30-year spin-up run was conducted by cycling meteorological forcing that was used for the actual simulation to obtain equilibrium for the soil water and soil heat balances.*

Page 5.Line 149. I would state that they were calculated from "observed data" rather than "model forcing". Are there any differences between the two?

*AR: We agree with the reviewer. This paragraph seems no need to be there anymore as the data processing method has been updated in section 2.2. We have deleted this paragraph.*

Page 5. Line 165-167. In my opinion, this classification of soil moisture conditions is very arbitrary, since there is no explicit link between the thresholds of SMI and plant physiologically meaningful variables such as "soil water potential" or better "leaf water potential". While overall, it is clear that with decreasing SMI drought stress should increase, there is no reason to support that drought stress should start at SMI of 0.1 or 0.4. I would suggest avoiding such classification and just having a continuous variable SMI.

*AR: We have renamed the groups of SMI to described soil moisture conditions (very dry: $0 \leq SMI < 0.2$, moderate dry: $0.2 \leq SMI < 0.4$, mid-range: $0.4 \leq SMI < 0.6$, moderate wet: $0.6 \leq SMI < 0.8$, very wet: $0.8 \leq SMI < 1$) rather than drought conditions (severe drought: $0 \leq SMI < 0.2$, moderate drought: $0.2 \leq SMI < 0.4$, mid-range: $0.4 \leq SMI < 0.6$, moderate wet: $0.6 \leq SMI < 0.8$, very wet: $0.8 \leq SMI < 1$). In the revised manuscript, we used continuous color bar for SMI in the figures. However, we still need the SMI groups for fittings to describe the responses of GPP and ET to environmental variables under different soil moisture conditions.*

Page 5. Line 171. I do not fully understand the rationale of using θsat in place of θfc. Even leaving a part the problematic concept of θfc (e.g. Assouline and Or 2014), the two values may be quite different and they are not interchangeable.

*AR: This is a matter of introducing an offset. We were not meaning the two parameters are changeable. As $\theta_{FC}$ acts as a proxy for $\theta_{SAT}$ in the JSABCH model technically (Hagemann and Stacke, 2015), for consistency, the $\theta_{SAT}$ was used instead of $\theta_{FC}$ when calculating SMI based on the observed soil moisture data. By doing this, the SMI still indicates the soil moisture conditions and simulated SMI and observed SMI are comparable.*

Page 5. Line 173-174. Also for Sodankylä, I 1do not understand why the authors did not use a soil water retention curve to link soil water contents to soil water potentials and therefore define some plant-meaningful threshold rather than arbitrary values.

*AR: This is a very good suggestion and worth trying. However, we have decided to leave Sodankylä out and mainly focus on Hyytiälä site, as there was no severe drought happened in our study period at Sodankylä.*

Page 6. Line 184. It is a measure of the inverse of the surface conductance.

*AR: It is a measure of the surface conductance, but not inverse. Our original formulation was correct.*

Page 6. Line 208-209. "deviated group" is awkward wording.

*AR: It has been deleted.*

Page 7. Line 235-236. As a follow up of my previous comment, SMI<0.2 does not necessarily imply a "drought" from the plant point of view and therefore the lack of deviation in GPP, ET, with respect to normal conditions should be expected as a consequence of the "lack of drought".

*AR: Yes, we agree with the referee. Because there is no severe drought in the study period, we decided to not include the Sondankylä site anymore.*

Page 8. Line 250-251. How many days are those for which we see such a deviation? Are they continuous in time? This is probably one of the most important information to place in the article, which needs to be emphasized.

*AR: We agree with the referee. In the revised manuscript, we have emphasized and gave detailed information about the severe soil moisture drought in 2006 at Hyytiälä in section 3.1: "The SMI in the summer of 2006 showed a period with SMI evidently lower (< 0.2) than in other years during the 11-year study period. According to the in situ observed SMI, in the summer of 2006, there were 37 consecutive days (23 July to 28 August) with SMI lower than 0.2, and 17 consecutive days (1 August to 17 August) with SMI lower than 0.15. The lowest SMI from observation was 0.115 on 16 August 2006. The simulated SMI was generally smaller than the observed SMI in the summer of 2006, showing 42 consecutive days (17 July to 27 August) with SMI lower than 0.2, and 33 consecutive days (26 July to 27 August) with SMI lower than 0.15. The lowest SMI from simulation was 0.052 on 15 August 2006."*

Page 8. Line 262-263. This is likely a consequence of the cross-correlation between high SMI and unfavorable meteorological conditions (e.g., cold, cloudy).

*AR: Yes, we agree this could be the reason. We have added this in the discussion.*

Page 9. Line 291-294. What is driving this proportionally larger decrease in GPP than ET? Looking from a leaf-level perspective this is hard to reconcile. What is keep sustaining ET if soil moisture is very low? How much is data uncertainty playing a role in such a response? Is it really just the change in VPD?

*AR: The low soil moisture leads to stomatal closure, and therefore decreased GPP and Transpiration. The larger decrease in GPP than in ET at the ecosystem level was because that the increased atmospheric demand of water (VPD) stimulated Evaporation from soil. The reason has been discussed in the discussion part.*

Page 11. Line 361-365. I agree that is important to highlight uncertainties of EC data, but how do you tackle this problem in the result presentation? There is not any confidence bound around the data, therefore we cannot really establish if some of data-driven result are very robust or not. At the very least, there should be a more quantitative discussion of the uncertainties.

*AR: It is difficult to provide any confidence bound in the figures. Instead, we could add some general discussion related to the uncertainty of EC flux data which is typically 20-30% for annual carbon budget Baldocchi (2003) and Aubinet et al. (2012).*

*Aubinet, M., Vesala, T., and Papale, D.: Eddy covariance: a practical guide to measurement and data analysis, Springer Science & Business Media, Netherlands, 2012.*

*Baldocchi, D. D.: Assessing the eddy covariance technique for evaluating carbon dioxide exchange rates of ecosystems: past, present and future. Global Change Biology, 9, 479–492, 2003.*

**Response to Anonymous Referee #5**

**General :**

The general topic of the manuscript falls well within the scope of the journal. The manuscript presents an interesting study where the authors used eddy covariance (EC) method and a land surface model

(JSBACH) to assess the response of daily water use efficiency (WUE; both ecosystem and intrinsic) to drought in two boreal forest ecosystems in Finland. The manuscript is well-written but may benefit from careful editing. The manuscript contains some interesting results, and I recommend addressing some comments and suggestion raised below to make it clearer.

**Specific comments:**

Was nighttime data included in the analysis? Obviously, there is no assimilation during the night and the EC system may produce or the LSM may simulate negative ET values. This may not be easily detected if daily flux sums, as opposed to sub-daily fluxes, were used but the computed WUE values would be fundamentally flawed and have no meaning. Some of the simulated results still contain negative ET values but no corresponding WUE (IWUE) values are shown for such data in the graphs. I recommend that this issue be addressed in the manuscript.

*AR: Agreed. The nighttime data were included in the analysis in the discussion manuscript. In the revised manuscript, according to reviewers' comments, only half-hourly data with shortwave radiation (Rs) larger than 100 W/m² were selected for the aim to select the effective time for plant photosynthesis. By doing this, data with negative GPP and ET were also excluded. The data selection process is described in section 2.2.*

It is not clear how the authors treated data during rainy days. Rainy days may underestimate WUE as evaporation from the canopy surface would be high during such times. It is vital that the conclusions drawn in this study do not include such low WUE values.

*AR: Agreed. In the revised data processing, rainy days and certain amount of dry days after the rainy days were excluded. The data selection process is described in section 2.2.*

Lines 109–114: Could the authors provide information as what percentage of the data was observed and/or gap-filled for CO2 and H2O fluxes. The authors stated that only good quality gap-filled data were used in the analysis. It is not clear whether data were discarded at half-hourly or daily time scale; or what percentage of 'good quality gap-filled data' was regarded good enough to be used in the analysis.

*AR: The original sentence was confusing and not totally correct. It has been deleted in the revised manuscript. Actually, the bad quality data was gap-filled, and the gap-filled data were used. The gap filling method for GPP and ET was introduced in section 2.2 in the manuscript. The percentage of gap-filled half-hourly $CO_2$ (for deriving GPP) is 35.8%, and the percentage of gap-filled half-hourly latent heat flux (for deriving ET) is 44.4%.*

Lines 115–118 and lines 147–150: Were these (Ta, Rs and VPD) daily averages? If so please specify. Also in all the graphs where these weather variables are used. I would prefer total daily Rs to daily average Rs values though. Do observations and simulations use the same weather variables (I would assume so)? However, I see quite a number of 0 W m-2 Rs observations (at Hyytiälä which is south of Sodankylä!) but not so many in the simulations. These could be because the corresponding simulated y-axis variables were 0 as well. The question is how possible is it to have daily Rs of 0 W m-2 in the summer.

*AR: Due to the new data processing for the revised manuscript, the Ta, Rs and VPD of the selected half hours in the selected days were averaged to represent the daytime mean Ta, Rs and VPD that effective*

*for plant functioning. The daytime mean Ta, Rs and VPD have been specified in the text and figure captions. For consistency with Ta and VPD, having just Rs per day does not make sense. We realized that the 0 W/m² Rs in the observational data is missing or bad quality data. However, those bad quality Rs has been gap-filled as the model forcing. In the revised manuscript, we selected the time periods with half-hourly Rs larger than 100 W/m² according to observational data for both observed flux and simulated flux. Thus the problem is not exist anymore in the revised manuscript.*

Lines 119–122. Time Domain Reflectometry (not Reflection). Also theta probe is not a technique, it is a sensor. It would be better if you mention the names (models) of the sensors and manufacturers, and readers could find out the techniques should they be interested.

*AR: The soil water content at Hyytiälä was measured at 1-h intervals by the TDR-method (Tektronix 1502 C cable radar, Tektronix Inc., Redmond, USA) connected to a data logger (Campbell 21X, Campbell Scientific Ltd, Leics., UK) via multiplexers (SDMX50, Campbell Scientific Ltd, Leics., UK). We are not using Sodankylä site for our study in the revised manuscript.*

As the soil water content forms the core of the simulation results, it would be helpful if the authors could provide more information on sensor types, how they were installed (vertically, horizontally or make use of access tube), if they were static or portable, measurement (time) interval, replications at a site, etc. Besides, was the soil water content for each sensor and depth calibrated against standard gravimetric measurements?

*AR: At Hyytiälä site, instrumentation for soil moisture was installed horizontally in the vertical face of five soil pits for each soil horizon (humus at 3 cm depth, the eluvial horizon at 8 cm depth, the illuvial horizon at 19 cm depth and parent material at 55 cm depth) 2 years before measurements were taken. The instruments were installed in the middle of each soil horizon, in the undisturbed soil at 20±30 cm distance from the face of the pit. Samples for measuring soil physical and chemical properties were collected from the walls of the pits during the excavation. The pits were filled with the original soil keeping the soil layers in the original order of excavation. We study the soil water content in mineral soil layers with the humus layer excluded. In the soil moisture data, the depth of mineral soil layers starts from 0 at the mineral soil surface.*

Why was the 0 to -5-cm soil water content at Hyytiälä disregarded in the observed vs simulated comparisons?

*AR: Because the soil moisture at the top mineral soil (i.e., layer-1) is too sensitive to climate varibility, which is not representative to show the soil moisture dynamics in the root zone. This is explained in the text in the revised manuscript.*

Lines 161–167: Again, it is not clear why observed and simulated SMI for layer 1 were not used, at least for Hyytiälä. And, how could the observed and simulated SMI be compared when the depth of the third layer is different for the two (at Hyytiälä)?

*AR: The 1st layer of the measured and the simulated soil moisture were not adopted because there is too much variability of the surface soil moisture in response to climate variability. This has been explained in the text in the revised manuscript. We selected the second and the third observed layers to cover the root depth. This has been correlated with simulations to select the best correspondence. The difference remains to some degree. In the revised manuscript section 3.1, the simulated SMI that calculated with simulated soil moisture and soil parameters in JSBACH has been compared with the observed SMI that*

*based on observed soil moisture and measured soil parameters. The simulated SMI agreed well with in situ observed SMI over the 11-year study period, with a correlation coefficient (0.625) and a root-mean-square error (RMSE) of 0.225. Moreover, a very good time correlation (0.965) between simulated and observed SMIs were found for year 2006, despite the simulated SMI is systematically lower than the observed SMI (RMSE = 0.12).*

165–179: Lack of measured soil parameters was the reason behind adopting different soil moisture classification units for the two locations. If there was continuous long-term soil water content data for so many years, it would not have been so difficult to derive these soil parameters (at Sodankylä). Saturated and field capacity soil water content may be estimated in spring following snowmelt, and wilting point from extended dry periods later in summer.

*AR: We agree with the reviewer's opinion. It would be good to estimate the soil parameters from long-time series data. However, the whole data series which is 8-year (2001-2008) data for Sodankylä site were used. Nevertheless, since there was no severe drought in the 8-year study period at Sodankylä, we decided to not include this site in the revised manuscript but mainly focus on the severe drought and its impacts at Hyytiälä site.*

Assuming the soils are similar at the two sites, it seems unlikely that the very wet conditions at Sodankylä would only have a soil water content of 13 to 16 .

*AR: The soil type is podzol at both sites. However, the mineral soil in Hyytiälä (the composition of mineral soil in 0-22 cm: clay 6%, silt 29%, sand 38%, stones 27%) is much finer than Sodankylä (the composition of mineral soil in 0-20 cm: clay 0.4%, silt 5.1%, sand 90.8%, stones 3.8%). This explains why the volumetric soil water content is much lower in Sodankylä.*

Lines 194–199: Could the authors provide R2 and P values for the graphs? What was the improvement in R2 and P values due to adopting exponential rather than linear relationship between GPP and ET or T?

*AR: The reason for adopting exponential rather than linear relationship is that more physiological relationships than just linear were used when existing. The R2 and P values for the relationships between GPP, ET or T and environmental variables (Rs, Ta and VPD) are provided in the table in the supplementary of revised manuscript.*

Lines 205–208: In general, there are more points in zone (b) than in zone (a); and zone (b) includes a larger range of the environmental variables. Both zones share more common than different environmental variables. I wonder if the environmental variables could be used at all to tease out differences in drivers between the two zones (apart from SMI).

*AR: This is exactly what we wanted to present that the zone (a) and zone (b) share more common environmental variables but have different soil moisture conditions. Moreover, in the revised manuscript, due to the new process of data, a group of data under low soil moisture content (encircled with a dashed line in Fig. 2(a)) showing GPP values lower than other days. The ET values of this group are also located in the lower end, but just partly lower than ET values on other days. It is found that the days in this group are all with very low soil moisture condition (SMI < 0.15). The old group (a) is not exist anymore with the exclusion of rainy days.*

Line 222: There are negative simulated ET values (Figs. S1, S2, S5, etc.). Could the authors explain

how they treated such data when determining WUE?

*AR: Those negative simulated ET values were dampened by daily averaging due to nighttime condensation. Through the reprocessed of the data for the analysis, only daytime data without precipitation influence were selected. The data selection process is described in section 2.2. In the revised manuscript, only half-hourly data with shortwave radiation (Rs) larger than 100 W/m$^2$ were selected for the aim to select the effective time for plant photosynthesis. The rainy days and certain amount of dry days after the rainy days were also excluded. By doing this, data with negative GPP and ET were excluded. Therefore, there is no problem of the influence from negative ET on WUE anymore.*

Lines 294–297: It would be better if the y-axis of Figs. 4 and S5 were IWUE rather than GPP×VPD. As it is, it is difficult to tell whether IWUE increases or decreases with ET.

*AR: The increase or decrease of IWUE can be seen from the ratio of GPP×VPD to ET, as the definition of IWUE.*

Be consistent in using either soil water content or soil moisture throughout.

*AR: We will use soil moisture throughout the text.*

Line 18 to 20: This is not clear at all. Rewrite please.

*AR: We have revised this sentence.*

Line 58: Change ". . . the ratio of GPP and ET." to ". . . the ratio of GPP to ET."

*AR: Agreed. It has been changed.*

Lines 80 to 83: Does not read well. Consider rewording–for example to ". . . the various land ecosystem model simulations highlight the current uncertainty with regard to plant physiology (water use) in response to drought."?

*AR: We have revised the sentence as: "The various land ecosystem model simulations highlight the current uncertainty about plant physiology (water use) in response to drought in models (Huang et al., 2015; Jung et al., 2007)."*

Lines 91 and 92: Consider removing this sentence and start with the actual experimental sites.

*AR: Agreed. This part has been rewritten for the Hyytiälä site.*

Line 93: 'annual temperature' should read 'annual air temperature'. Also in Table 1 (Lines 600–605).

*AR: Agreed. It has been changed.*

Line 116: Humidity is ambiguous term to use. Be specific which measure of atmospheric humidity was used.

*AR: Agreed. It has been changed.*

Line 153: Delete the phrase 'to be able'

*AR: We have deleted the "to be able".*

Line 157: "... ] and θWILT is the ..."

*AR: This has been changed according to the comment.*

Line 170: "... (i.e., soil water content ..."

*AR: This has been changed according to the comment.*

Line 170: "... and θWILT= ...)"

*AR: This has been changed according to the comment.*

Line 171–172: Confusing statement. This changes the definition of SMI and probably the boundaries of SMI set-up above. And, if θFC acts as a proxy for θSAT, then should not θFC be used instead of θSAT?

*AR: Because $\theta_{FC}$ plays the role as $\theta_{SAT}$ in the JSABCH model technically (Hagemann and Stacke, 2015), for consistency, the $\theta_{SAT}$ was used instead of $\theta_{FC}$ when calculating SMI based on the observed soil moisture data. By doing this, the SMI still indicates the soil moisture conditions, and simulated SMI and observed SMI are comparable. Using $\theta_{FC}$ or $\theta_{SAT}$ is a matter of introducing an offset.*

Line 192 onwards: ET/T is ambiguous. Use ET or T instead.

*AR: This has been changed according to the reviewer's suggestion.*

Line 248: Replace 'slop' with 'slope' in Tables S1 and S2.

*AR: Indeed. It has been changed.*

Line 300: Simulated EWUE is not presented in Fig. S5.

*AR: The simulated EWUE is now shown in the supplementary figure.*

Lines 304–305: This sentence is not a result. Move it to the Discussion Section.

*AR: We agree with the reviewer. However, this part about Sodankylä results has been deleted in the revised manuscript.*

Lines 313, 320 and 337: Consider changing the word 'disturbance'. Some suggestions:
Line 313: 'Moreover, GPP and ET were decoupled and EWUE decreased ...', Line 320: '... there was no deviation in GPP, ...', and Line 337: 'The simulated daily ET data contained frequent ...'.

*AR: We have revised the sentence in Line 313. The other two sentences are not exit in the revised manuscript due to the exclusion of Sodankylä site in the revision.*

Line 341: '... limitations on GPP and ET or T relationships under ...'.

*AR: We do not agree to add "relationships" in the sentence as we referred to the impacts of drought on GPP and ET.*

Line 344: '. . . when soil moisture was under . . .'.

*AR: We agree with the reviewer about this grammar mistake. The sentence has been modified.*

Line 370: Delete 'as a whole'

*AR: Yes, this has been deleted.*

---

## Author Response (AR2)

**point-by-point response to the reviews**

**Reply to Anonymous Referee #5**

**General comments:**

The authors did a good job improving over the original submission. They dealt well with most of the comments raised on earlier review. However, the manuscript contains many language errors and I recommend that it carefully be edited by a native English speaker. Some are listed below in the minor comments but the list is by no means exhaustive. Also where graphs share the same y-axis title and y-axis label, the readability of the graphs could be enhanced by using only one (on the left hand side) y-axis title and y-axis label instead of having them for each panel (for example, Figs. 2 and 3).

I recommend accept with minor revision.

*Authors response (AR): We appreciate the reviewer's detailed and helpful comments. In the revised version, we have tried to correct language errors listed below and in some other places. The graphs sharing the same y-axis title and y-axis label have been replotted, using only one y-axis title and y-axis label.*

**Minor revisions to consider:**

Lines 71–72: Not sure what message this sentence is meant to convey?

*AR: We have reformulated this sentence as "Beer et al. (2009) concluded that the impact of vapour pressure deficit (VPD) on canopy conductance disturbs responses of both GPP and ET to changing environmental conditions and proposed the ecosystem level inherent water use efficiency (IWUE), which is a quantity defined as EWUE multiplied with mean daytime VPD." to introduce IWUE.*

Lines 76–77: 'The appropriateness of uWUE at daily time scale was also demonstrated (Zhou et al., 2015).'??

*AR: We have revised the sentence according to reviewer's suggestion.*

Lines 87–88: ??… (3) to evaluate how adequately the JSBACH land surface model (LSM) captures plant responses to changes in environmental variables.??

*AR: We have modified this sentence according to reviewer's suggestion.*

Line 95: Check LAI in Table 1.

*AR: Yes, we have checked the LAI values in Table 1. There was no mistake of the LAI values in Table 1. To clarify, we added explanation about the maximum LAI set in JSBACH in the manuscript as " The modelled LAI reached values close to the observed LAI when the parameter maximum LAI was set to $16 \ m^2/m^2$".*

Line 96: '+' is not necessary.

*AR: We have deleted '+'.*

Line 107: ' … by excluding …'

*AR: We have added 'by' in here.*

Lines 120–121: Move this to line 193 to explain why the first layer was not included in the analysis.

*AR: We have added this reason in line 193.*

Line 215: Perhaps present some measure of variation (e.g. standard deviation, confidence interval, etc) in addition to averages in the 11-year climatic conditions so that readers could have an idea of the deviations during the drought period.

*AR: We have provided the standard deviation in addition to averages in the 11-year climate conditions in the text.*

Lines 222 and 224: Coefficient of determination was used elsewhere; no need to put the numbers in brackets; is three significant digits necessary here?; what is the difference between correlation coefficient and time correlation?.

*AR: We have used correlation coefficient in all the places in the revised manuscript. We agree that there is no need to put the numbers in brackets and we have changed this. We changed three significant digits for correlation coefficients to two significant digits. There was no difference between correlation coefficient and time correlation in what we were trying to mean there. For clarity, we changed time correlation to correlation coefficient.*

Line 225: Replace 'was' with 'being'

*AR: 'was' have been replaced with 'being'.*

Line 228: Replace 'presented' with 'observed'

*AR: Yes, we have replaced 'presented' with 'observed'.*

Line 230: Replace 'is' with 'was'

*AR: This error has been corrected.*

Line 232–233: Please reword the sentence 'The biggest difference of 1.054 kPa took place on August 5th, the day with highest Ta in August 2006.'

*AR: We have reworded the sentence as "The biggest difference between the daily mean VPD and the 11-year averaged daily mean VPD reached 1.054 kPa on August $5^{th}$ that was the day with highest $T_a$ in August 2006." to make it more clear.*

Line 266: Replace 'environment' with 'environmental'

*AR: We have replaced 'environment' with 'environmental'.*

Lines 291–292: Provide units; the SMI values for the IWUE values are swapped??

*AR: We have provided the units for IWUE here and corrected the mistake on the swapped SMI values.*

**Reply to Referee #6**

The authors responded well to the comments arisen. A minor concern, I suggest arranging the content of Discussion and Conclusion sections. I prefer a brief and clear conclusion with the key finding of the study, however, the current Conclusion is uneasy for the readers to understand the highlights. I think the related discussion on mechanism of impacts of drought on different WUE types should be embedded into the Discussion sections.

*Authors response (AR): We appreciate the comments from the reviewer. In the revised version, we have tried to rewrite the conclusion to be more focused on the key findings of the study. We hope the current conclusion is easy for the readers to understand the highlights.*

**Reply to Referee #7**

In the first round of the review, the reviewers have addressed some important points (e.g. filtering of the data, focus on one site only, classification of SMI values etc.).
Most of these comments were adequately addressed by the authors, and the revised version of the manuscript has improved in many aspects. The Methods and Results sections are clearly and well written.
However, the discussion is weak and lacks novel aspects. In particular, I do not agree with the conclusion of IWUE being the most appropriate WUE metric, and I doubt that this question can be answered based on the results of this study. See more details below.

*Authors response (AR): We appreciate the reviewer's constructive comments. In the current manuscript, we have revised the discussion and conclusion about water use efficiency metrics. Please see the details in the revised manuscript. We hope our revision solves the reviewer's concern.*

**More detailed comments:**

- I think the introduction would be stronger if the authors mentioned the anticipated occurrence of drought in this region (e.g. after line 49). In the current version, this is only mentioned in the conclusions.

*AR: We have added the anticipated occurrence of drought in this region in the revised manuscript.*

- line 74: "simple stomatal model" does not adequately describe the model by Lloyd & Farquhar 1994. I suggest to reformulate this in a way that clarifies that it is based on the optimality theory developed by Cowan & Farquhar 1977.

*AR: Yes, we have reformulated this sentence as "an optimality theory (Cowan and Farquhar, 1977) based stomatal model with the assumptions suggested by Farquhar et al. (1993) and Lloyd and Farquhar (1994)".*

- I agree not to use layer 1 soil moisture, but the justification "too sensitive to climate variability" is a bit misleading as "climate" implies a longer time scale than what you mean here.

*AR: We will modify the "climate variability" to "temperature and precipitation variations".*

- line 165: replace "JSABCH" with "JSBACH"

*AR: We have corrected this error.*

- line 193f: not clear how these soil parameters were determined. Please explain shortly.

*AR: The soil parameters are derived based on water retention curves determined from soil samples taken at the site. We have added this explanation in the manuscript.*

- line 204f.: I would not introduce IWUE as a measure of surface conductance. This metric accounts for the confounding effects of VPD on ET, but it does not consider that also Gs decreases with increasing VPD (but uWUE does). Further, the derivation of IWUE is based on strong assumptions (e.g. that Ga is infinitely high) that are not justified when this metric is interpreted in terms of Gs. It's ok to use this metric, but the authors should not over-interpret its physiological meaning. I suggest to introduce it simply as a WUE metric that accounts for the effects of VPD on ET. For the same reasons, don't call IWUE "intrinsic WUE" in line 307

*AR: We agree with the reviewer. Thus, we have deleted the places that introduce IWUE as a measure of surface conductance at the ecosystem level and as a representative of intrinsic WUE at the ecosystem level.*

- line 207: cite Zhou et al. 2014 again here, who suggested this metric

*AR: We have added Zhou et al. 2014 here in the revised manuscript.*

- An interesting aspect of Figure 1 is that in some years (e.g. 2003, 2009), the simulated and observed soil water dynamics do not agree well compared to the other years. Do the authors have an explanation of why this could be the case?

*AR: Indeed, we agree with the reviewer that there are some discrepancies between the simulated and observed soil water dynamics in some years. The soil water dynamics in the model show faster response to both lack of water supply as well as to precipitation events than the observations. Furthermore, deviations in the initial values of soil moisture after the snow melt in the beginning of the season impacts the soil moisture in the later season. Differences in the dynamics are not surprising as the model does not account for the vertical layering of the podzol soils and the model parameters are not calibrated with the measurement data. Regardless of the deviations in the dynamics, the model was able to produce both the timing and the spatial extent of the 2006 drought (Gao 2006).*

- It would probably be too messy to show VPD in Figure 1 as well, but it would help to provide the absolute range of VPD observed in the drought period in the text, and not just the deviations compared to the reference period.

*AR: We have added the mean value and standard deviation of VPD during the drought period in the*

*text.*

- Lines 255f: this sentence is a bit unclear. Isn't it the other way around? I.e. the GPP-ET relationship is more linear in the simulations?

*AR: We think the GPP-ET relationships are non-linear in both observation and simulation. To make it more clear, we have reformulated this sentence as "The non-linear relationship between the daytime averaged GPP and ET was also found in the JSBACH simulated result".*

- I do not agree with most parts of section 4.1 (and the corresponding sections of the conclusions). The authors try to identify the most appropriate WUE efficiency measure, but they do not really explain why IWUE is considered more appropriate than the other metrics. Is it just because IWUE increased, whereas uWUE does not show a response to drought? That may be true but how do we know that this is then the more appropriate response? Another aspect that was not mentioned is that uWUE in Fig. 4 has a better correlation than IWUE. Isn't that an indication that uWUE is a better metric describing the water-carbon coupling? In my opinion, this part of the discussion lacks some more fundamental considerations of the metrics applied. For instance, uWUE was developed to overcome some shortcomings of the IWUE metric (i.e. the stomatal response to VPD is taken into account), and in that sense is a more meaningful metric to characterize ecosystem functioning than IWUE. The weaker VPD response of IWUE should also (or in particular) play a role in drought periods, so I don't follow the argumentation why uWUE should be better suited for long-term studies than for particular drought periods.

*AR: We agree with the reviewer's comments. We revised the discussion in section 4.1 and concluded that the increase in IWUE during drought was a result of decreased stomatal conductance due to increased VPD. The uWUE was formulated to be more independent of a varying VPD than IWUE. The unchanged uWUE during this short-term drought demonstrated that the trade-off between carbon assimilation and water transpiration of the plants at this site was not disturbed by drought, even though the stomatal conductance decreased. We also agree with the reviewer that it is incorrect to state that uWUE is better suited to long-term studies where individual drought periods are not on the focus.*

- other main points of the discussion are rather weak and repeat outcomes of earlier studies. For example, it is long known that stomatal conductance responds to air humidity. It is also commonly accepted that non-stomatal limitations are needed to adequately simulate the drought response in models. These aspects are long known to the modelling community.

*AR: We agree that it is ideal that ecosystem models can have a very good representation of stomatal functioning and its dependence on VPD. Even though it is long known that stomatal conductance responses to air humidity and the non-stomatal limitation are needed, in global ecosystem models, simple representations of stomatal regulation have often been applied to reduce computing costs. Because VPD and soil moisture are to certain degree correlated, inclusion of one of the either has often been considered to be enough to account for drought effects. The performance of the default stomatal conductance model in JSBACH under limited soil moisture conditions has not been tested before this study. Our results indicate that the combined effects of soil moisture and atmospheric drought on stomatal conductance have to be both taken into account.*

- Line 318: rewrite to "stomatal conductance in JSBACH does not include a response to air humidity" or simply delete the first part of the sentence.

*AR: We have modified this sentence according to reviewer's suggestion.*

- Figure 2a: Explain the meaning of the dashed circle also in the caption, not only in the text

*AR: We have added the meaning of the dashed circle in the figure caption.*

- Wouldn't it make more sense to present GPP in gC m-2 d-1? This would also be easier to interpret. Up to the authors.

*AR: We agree that to present GPP in gC $m^{-2}$ $d^{-1}$ is easier to interpret. However, we selected the day time periods of effective photosynthesis with the criteria that incoming shortwave radiation (Rs) larger than 100 W/$m^2$. This means that we selected different time periods of different days. Thus, we think it is more proper here to use mean GPP and ET rates instead of daily sums here.*

- Table 1: It would make sense to indicate the grain size distribution under "Soil type" in the lower part of the table.

*AR: We tried to select the most reasonable soil type for the site level simulation to match with the soil type at the site. Nevertheless, there is no perfect match as we can only chose one soil type for the site in the model rather than mixed soil type in the reality. Unfortunately, there is no information on grain size distribution for the FAO soil types in the five layer soil hydrology scheme in JSBACH, which has been developed in Hagemann and Stacke (2015). There are only soil parameter values for the different FAO soil types which are taken from various sources. Interested reader can check those soil parameters for FAO soil types used 
[revised manuscript text omitted]

755

| | |
|---|---|
| **Deleted:** − |
| **Deleted:** − |
| **Deleted:** − |
| **Deleted:** − |